# The Functional Significance of Hydrophobic Residue Distribution in Bacterial Beta-Barrel Transmembrane Proteins

**DOI:** 10.3390/membranes11080580

**Published:** 2021-07-30

**Authors:** Irena Roterman, Katarzyna Stapor, Piotr Fabian, Leszek Konieczny

**Affiliations:** 1Department of Bioinformatics and Telemedicine, Jagiellonian University Medical College, Medyczna 7, 30-688 Kraków, Poland; 2Institute of Computer Science, Silesian University of Technology, Akademicka 16, 44-100 Gliwice, Poland; katarzyna.stapor@polsl.pl (K.S.); piotr.fabian@polsl.pl (P.F.); 3Medical College, Jagiellonian University, Kopernika 7, 31-034 Kraków, Poland; mbkoniec@cyf-kr.edu.pl

**Keywords:** transmembrane proteins, hydrophobicity, hydrophobic core, periplasmic, oil transport, antibiotic resistance, transport channels

## Abstract

β-barrel membrane proteins have several important biological functions, including transporting water and solutes across the membrane. They are active in the highly hydrophobic environment of the lipid membrane, as opposed to soluble proteins, which function in a more polar, aqueous environment. Globular soluble proteins typically have a hydrophobic core and a polar surface that interacts favorably with water. In the fuzzy oil drop (FOD) model, this distribution is represented by the 3D Gauss function (3DG). In contrast, membrane proteins expose hydrophobic residues on the surface, and, in the case of ion channels, the polar residues face inwards towards a central pore. The distribution of hydrophobic residues in membrane proteins can be characterized by means of 1–3DG, a complementary 3D Gauss function. Such an analysis was carried out on the transmembrane proteins of bacteria, which, despite the considerable similarities of their super-secondary structure (β-barrel), have highly differentiated properties in terms of stabilization based on hydrophobic interactions. The biological activity and substrate specificity of these proteins are determined by the distribution of the polar and nonpolar amino acids. The present analysis allowed us to compare the ways in which the different proteins interact with antibiotics and helped us understand their relative importance in the development of the resistance mechanism. We showed that beta barrel membrane proteins with a hydrophobic core interact less strongly with the molecules they transport.

## 1. Introduction

β-barrel transmembrane proteins are expressed in the outer membrane of bacterial cells. Since mutations in these proteins have been implicated in conferring antibiotic resistance, knowledge of their structures and functions is important for the development of clinical therapies [1,2,3]. The prevalence of drug resistance, especially in bacteria strains observed in hospital-acquired infections, have important implications for modern medicine [4,5,6,7,8,9,10,11,12,13,14]. New solutions for overcoming resistance, which commonly arises from the misuse and over-prescription of antibiotics, are required [4,15,16].

At the present stage in the development of therapeutic techniques for which the introduction of personalized medicine methods poses a new challenge, the search for new solutions is becoming a necessity [4,17,18]. Systems’ biology approaches that integrate the knowledge from structural and functional studies of transport proteins and related signaling molecules must be employed in the development of new antibiotics [19,20].

To enter the cell of a Gram-negative bacterium, an antibiotic must penetrate the outer membrane. A molecular understanding of how drug molecules pass through the channels in this membrane is essential when endeavoring to develop new therapeutic compounds [21,22,23,24,25]. The analysis of this phenomenon at the molecular level focuses on the mechanism involved in the penetration of the designed antibiotic into the transmembrane channels. Proteins from the group, known as OmpX (outer membrane protein X), play an essential role at this stage. These proteins are located in the outer membrane of the bacteria and connect the periplasm with the outside world. The subject of the current analysis is a set of bacterial proteins from the OmpX group. We focused on their structures, characterized by the presence of regular beta-barrels, analyzing their adaptation to the hydrophobic environment of the membrane and preparation for the transport of various molecules. Our analysis is based on the use of the fuzzy oil drop (FOD) model and, in particular, the modified version of this model—the FOD-M model, where M expresses the membrane environment. With the help of this analysis, it is possible to explain the specific features of the proteins in question, such as their differentiated resistance to some forms of antibiotics and their specificity in relation to the transported molecule, such as oil transport through a membrane.

The aim of the study is to demonstrate the applicability of the fuzzy oil drop model as a tool for analyzing transmembrane proteins that act as transport channels not only for the various compounds necessary for the life of bacteria but also for potential drugs in the form of antibiotics. The set of proteins in question include examples of various structures, structural forms, and the role of bacteria, although proteins with the helical transmembrane part are discussed in reference [26]. Here, only one representative of this group is discussed. The fuzzy oil drop model has been proven to be a suitable platform for the evaluation and structural–functional characterization of transmembrane proteins.

## 2. Materials and Methods

### 2.1. Data

Table 1 shows the membrane proteins of the OmpX group that act as channels for various molecules and are the object of the present analysis. The analysis of these proteins was performed using a modified form of the fuzzy oil drop model (FOD-M), for which hydrophobicity distribution is the key criterion.

### 2.2. Fuzzy Oil Drop Model and Its Modification Taking into Account the Influence of Factors Other Than the Aqueous Environment

Both the fuzzy oil drop model (FOD) and its modified form (FOD-M), which takes into account the influence of the nonpolar environment on the membrane protein structure, have been presented in numerous papers [26,39,40,41]. The short model description provided here is intended to facilitate the interpretation of the presented results. The FOD model uses a 3D Gaussian function to describe the distribution of hydrophobic amino acid residues in globular proteins. The value of this function (spread over the protein body and expressed by appropriately selected parameters sigmaX, sigmaY, and sigmaZ) at the position of the effective atom (averaged position of atoms belonging to a given amino acid) determines the idealized level of hydrophobicity for a given amino acid, designated as *Ti*.

At the same time, the value of the actual hydrophobicity level resulting from hydrophobic interactions (depending on the distance between the effective atoms and the intrinsic hydrophobicity of the interacting amino acids) is determined—*Oi* [42] The analysis also takes into account the *R* distribution, which is the opposite of the *T* distribution. It represents a state in which each residue is assigned an identical level of hydrophobicity = 1/*N*, where *N* is the number of residues in the chain. Such a distribution presupposes a uniform distribution of hydrophobicity throughout the protein body without identifying the hydrophobic core in particular. Normalization of all the discussed distributions (*T*, *O* and *R*) enables a comparison of them by means of divergence entropy—*D_KL_* [43]:(1)$$D_{KL}(P|Q)=\sum_{i=1}^{N}P_{i}\log_{2}\frac{P_{i}}{Q_{i}}$$
where *P* expresses the analyzed distribution—in our case, the *O* distribution—*Q*, the reference distribution, and, in our case, the *T* or *R* distribution.

The application of the above definition to the *T*, *O* and *R* distributions results in the following formula:$$D_{KL}(O|T)=\sum_{i=1}^{N}O_{i}\log_{2}\frac{O_{i}}{T_{i}}$$

The *D**_KL_* value thus determined expresses the distance between the idealized distribution and the distribution observed in a given protein. On the other hand, the distance between the *O* and the *R* distributions is calculated as follows:$$D_{KL}(O|R)=\sum_{i=1}^{N}O_{i}\log_{2}\frac{O_{i}}{R_{i}}$$

In this formula, the reference distribution is the R distribution representing the state with a uniform distribution of hydrophobicity throughout the protein’s body.

The *D_KL_* value expresses the distance between the two compared distributions on the entropy scale. As a consequence, the final analysis is based on a comparison of *D_KL_* for the *O|T* and the *O|R* relations, respectively. The relation *D_KL_ (O|T)* < *D_KL_ (O|R)* suggests a compatibility of the *O* and *T* distributions, which is interpreted as the presence of a hydrophobic core with the simultaneous exposure of hydrophilic residues on the surface (in accordance with the Gaussian function characteristics). To eliminate the need for two quantities, the parameter *RD* (Relative Distance) was introduced, expressed as follows:(2)$$RD=\frac{D_{KL}(O|T)}{D_{KL}(O|T)+D_{KL}(O|R)}$$

An *RD* value <0.5 is interpreted as the presence of a hydrophobic core.

In addition to the polar (water) environment, proteins also function in the hydrophobic environment of the cell membrane. Here, the nonpolar character of the lipid environment is expected to promote the exposure of hydrophobic residues on the protein surface. Additionally, if the membrane protein acts as a channel, and especially if it acts as an ion channel, polar residues are expected to point inwards to stabilize a water-filled internal pore. Therefore, to describe the hydrophobicity distribution in the membrane proteins, the distribution expressed by the complement to the Gaussian function value is assumed to be *T_MAX_**–**Ti*, where *T_MAX_* is the maximum value obtained for the T distribution for a given protein.

Due to the universality of the aqueous environment, its influence is taken into account by introducing a certain consensus between the influence of the polar aqueous environment and the hydrophobic environment. As a result, the external field for the activity of the channel membrane protein is determined by the distribution of *M* in the form:*Mi* = [*Ti* + *K* ∗ (*T*_*MAX*_–*Ti*)_*n*_]_*n*_(3)
where index *n* denotes normalization.

The introduced parameter *K* expresses the participation of the hydrophobic environment, which was previously defined as an expression of the consensus between these two external fields: the polar field coming from water and the nonpolar field coming from the membrane. The value of the *K* parameter turns out to be specific for a given protein, characterizing the distinctiveness and diversity of the world of proteins.

The characteristics of the proteins discussed here depend on the value of the *RD* parameter for the *T**–**O**–**R* relationship, with the *T* and *R* distributions as reference distributions, as well as the value of the *K* parameter defining the participation of the nonpolar factors in the hydrophobicity distribution within a given protein. The *K* parameter determines the conditions of the consensus between the two environments: polar (water) and hydrophobic membranes, regardless of the factors causing it.

The identification of the *K* value for which the *D_KL_* value for the *O|M* relation (where *M* denotes an adequately modified *T* distribution) makes it possible to determine the status of a given protein in relation to the hydrophobic environment of the membrane. *K* values close to zero denote the predominant (or even the only) participation of the aqueous environment. *K* values close to 1 or even above 1 denote a significant participation of the hydrophobic environment, thereby creating the conditions for the biological activity of a given protein and probably the participation of nonpolar factors in shaping the protein structure. The value of the *K* parameter determined for a given protein determines the form of the external field in which a given structure is enfolded or in which the protein functions and demonstrates its biological activity.

Analogously to the relation of *T**–**O**–**R*, the parameter *RD* can be introduced for the relation of *T**–**O**–**M* and, based on *D**_KL_*_,_ determined for the distribution *M* treated as the reference distribution:$$D_{KL}(O|M)=\sum_{i=1}^{N}O_{i}\log_{2}\frac{O_{i}}{M_{i}}$$

The *RD* parameter for the relations of *T**–**O**–**M* can be calculated as:$$RD=\frac{D_{KL}(O|T)}{D_{KL}(O|T)+D_{KL}(O|M)}$$

It defines the degree of compatibility between the O distribution and the modified *M* distribution. The higher the value of *RD* for this relation, the closer the distribution *O* to the distribution *M*. The *M* distribution replaces the previous reference distribution *R*, which expresses the complete absence of hydrophobicity diversity, including the complete absence of the hydrophobic core. The degree of compatibility between the *O* and *M* distributions reveals the type of hydrophobicity diversity present in a given protein.

Figure 1 presents the relationships between the distributions.

## 3. Results

Table 2 summarizes the results of the FOD-M analysis for membranes of the OmpX beta barrel membrane family.

The items in Table 2 may be interpreted as follows.

The *RD* value for the *T**–**O**–**R* relationship determines the degree of hydrophobic core presence compared to a system completely devoid of this presence. *RD* values <0.5 for this relationship suggest the presence of a hydrophobic core.The *K* values indicate the degree of participation of a factor other than polar in shaping the structure. The higher the value of *K*, the greater the proportion of the factor different from the aqueous environment, including the hydrophobic environment of the membrane in particular.The *RD* values for the *M**–**O**–**T* relationship express the degree of adjustment of the *O* distribution to the modified T distribution called *M* with the reference T distribution. Low *RD* values for *M**–**O**–**T* indicate that *O* is “approaching” the modified distribution, where the degree of modification is expressed by the value of parameter *K*.The value of *K* is determined using the step-wise procedure for successive *K* values, which involves looking for the minimum *D_KL_* for the *O|M* relationship.

### 3.1. Proteins Implicated in Antibiotic Resistance (OprH and OmpA)

Both of these proteins show (contrary to what was predicted) a hydrophobicity distribution consistent with the 3D Gauss distribution, which can be interpreted as structures with a central hydrophobic core and polar residue exposure (Figure 2A). This solution is surprising for membrane proteins. Hydrophobic residue exposure in beta-barrel segments in contact with the membrane was expected instead (Figure 3). Moreover, despite the visible channel in the central part of the beta-barrel, the *T* and *O* profiles do not show a hydrophobicity deficit (i.e., the compatibility of hydrophobicity maxima). The beta-barrel status in the case of the 2LHF protein shows a high *RD* value for the *T**–**O**–**R* relationship, which suggests an arrangement different from the centric hydrophobic core. For this beta-barrel, an increased K value in relation to the 0.4 value occurs (the complete molecule is described as *K* = 0.2), which means the need to modify the form of the outer field for the beta-barrel itself. A different situation is observed in the case of 2JMM, where the status of the beta-barrel itself is even more consistent with the distribution featuring a centric hydrophobic core than in the case of the molecule as a whole. The experimental modification shortening the loop length introduced in 2JMM [28], the purpose of which was to examine the role of these loops in resistance phenomenon, did not change the status of this protein (according to the fuzzy oil drop model), with a hydrophobic core being present despite the channel in the central part of the molecule

Mutations in Protein H (OprH; PDB ID 2LHF) are implicated as a cause of antibiotic resistance in *Pseudomonas aeruginosa.* This bacterium is a major nosocomial pathogen that infects cystic fibrosis and immunocompromised patients [27]. An analysis of the structure of the hydrophobic core and the characteristics of the 2JMM structure appear to suggest and explain the higher resistance to binding of any molecule compared to 2LHF. The agreement of the *T* and *O* distributions in this protein is expressed by a high correlation coefficient of 0.969.

Based on the previous analyses, a structure with a hydrophobic centric core poorly interacts with other molecules. The binding site of the ligands (substrates) in proteins was identified in the profile as a local hydrophobicity deficit resulting from the presence of a polar cavity [44]. The secondary binding site for substrates is often associated with the hydrophobic region of the protein [45]. Complete coverage of the protein surface with polar groups eliminates the possibility of interactions with molecules other than water.

For example, antifreeze proteins show similar characteristics to the proteins discussed here. Antifreeze proteins that only interact with water show similar characteristics to the proteins discussed here [45]. This similarity suggests a very low probability that the proteins in question can interact outside an aqueous environment.

The distribution of polar and nonpolar residues in OprH and OmpA suggests that they interact weakly with the substrate molecules, including antibiotic molecules, which is consistent with previously published research evidence.

The fuzzy oil drop model does not explain the absence of any correlation between the hydrophobic residue resistance and the structure of the transmembrane beta-barrel. On the other hand, the very compatibility of the *T* and *O* distributions may explain the reason for resistance in the case of these proteins.

### 3.2. Oil Transport (6QAM, 6QWR)

The protein AlkL increases the permeability of the *Pseudomonas oleovorans* outer membrane for hydrophobic molecules. Two structures, one soluble in the presence of a lipid (6QWR) and one in the presence of a detergent (6QAM) were analyzed in this study to determine the importance of the composition of a hydrophobic environment. The proteins represent identical sequences that additionally support the investigation of the environment’s impact on a structure. Different hydrophobic environments (lipid and detergent) have been shown to influence the membrane protein structure [29], and AlkL provides an opportunity to test the application of the FOD-M analysis in two structures obtained using different NMR methodologies [30]. The status of the molecule as a whole seems to be comparable for both forms *(RD* and *K*), although the detergent environment produces a greater deviation than the T distribution. The status of the beta-barrels themselves is different from an *RD* point of view. However, the modification of the reference distribution at the level *K* = 0.4 in both cases results in a significantly increased similarity between the *O* and *M* distributions in the case of the lipid environment (*RD* for low *M**–**O**–**T* low), while, in the detergent *RD* environment for *M**–**O**–**T*, the similarity is much less (Figure 4). However, the status of the non-beta-barrel part is different. In the case of the detergent environment, the deformation of the starting field (3DG) was considerable (*K* = 1.2), while, in the case of lipids, this deformation was expressed with the value of *K* = 0.7. The modification expressed by the distribution of M in both cases was significant, leading to a decrease in the *RD* for *M**–**O**–**T* down to almost 0.3 and, in the case of the lipid environment, to a level of 0.2.

A comparison of the *T* and *O* profiles in both the proteins in question revealed the redundancy of hydrophobicity in the outer sections in relation to the expected level. This signified adaptation to the membrane environment. On the other hand, the expectations of the centric core were, so it seemed, partially realized. Most of the local expected peaks were only partially reconstructed. In addition to the clear adjustment of some local peaks, there was a significant deficit of about half the width of a given maximum (sections 15–30, 70–85, 160–175, or 185–200). A comparison of the *T* and *O* profiles with the *RD* and *K* values for the two structural forms of AlkL (6QWR lipid and the 6QAM detergent) present in the environment showed the stronger influence of the detergent on the structure of the system (Figure 5). The status of segment 30–40 is characteristic, which, in both cases, shows (as was expected in the case of the membrane proteins) a significant local excess of hydrophobicity on the surface. Section 115–130 shows a significantly higher level of hydrophobicity than was expected within this local maximum. This signifies its adaptation for the transport of a highly hydrophobic molecule, an interaction which, inside the channel, is probably beneficial.

The significant differences in the position of the out-barrel loops, visible in Figure 5, are reflected in the parameters calculated in this analysis, presented in Table 2.

### 3.3. Beta Barrel Proteins of Higher Diameter (OmpA, BamA)

The beta-barrel proteins described in this section are characterized by a much larger barrel cross-section diameter (Table 3) than those discussed previously. It turns out, however, that even in this group with a clearly high degree of similarity at the super-secondary structure level, differentiation can be observed. The summary includes a protein with a diameter comparable to those previously discussed. There are considerable variations in the hydrophobicity profiles of members of this group, as shown in the differences in the *K* values calculated in this analysis. The 1QJP structure of OmpA (average internal diameter = 15. 9 Å; *K* = 0.4) closely resembles other beta-barrel proteins with a clear hydrophobic core, whereas the 3K3C structure of BamA (average internal diameter = 35.2 Å; *K* = 1.2) has a more polar interior. Similarly, the status of the beta-barrel itself changes with K from 0.6 to 1.3. The beta-barrel status is expressed by a higher *RD (T**–**O**–**R)* value relative to the complete molecule. This results in an increasing fit of the *O* distribution to the *M* distribution (versus the reference T distribution). The proteins discussed here are classic examples of the so-called “inverted” field on the *K* = 1 scale. This means that the polar and hydrophobic environment equally influence the formation of this transmembrane beta-barrel. The T and O profiles show both the exposure of hydrophobic residues on the surface (increased levels of hydrophobicity in places of exposure) and lowered levels of hydrophobicity in those sections constituting channel components (Figure 6).

To visualize the differences in the barrel size characteristics of the discussed proteins, Table 3 presents the average diameters of the channels.

The summary presented in Table 3 shows the different values according to the size of the channel. The adaptation to the function performed becomes visible. A strong correlation exists between the sizes of the diameters and the sizes of the possible molecules or drug transport. The data given here will be discussed in the next parts of this paper.

The profile summary for the proteins discussed here reveals a gradual match with the expected distribution typical for a transmembrane protein. The *T* and *O* distributions in the case of OmpA (PDB: 1QJP) show relatively strong agreement (*RD* for *T**–**O**–**R* exceeding the cut-off level of 0.5) up to the level of *RD* = 0.726 for 4N75. An increase in the *O* values occurs in the sections of the protein exposed to the environment. A wide range of sections show a large deficit between the *T* and *O* maxima [46].

Differences can also be observed in the M distributions, which move ever closer to the straight line. The straight line parallel to the x-axis represents the R distribution. This distribution expresses the uniform distribution of the level of hydrophobicity throughout the protein body. This is interpreted as independence from environmental influences. This means that no environment (either polar to generate a centric hydrophobic core or hydrophobic, aiming at hydrophobicity exposure on the surface) has an effect on the molecule’s shape. The composition of the amino acid of such a molecule is an environmental field in itself. The presence of an R-type distribution in the protein is interpreted as constituting a field generated by the molecule itself, thus eliminating both effects of targeting. This is exactly the case with the 4N74 and 4K3C BamA structures. They are large molecules (379 and 532 amino acids, respectively). It is the number of molecules that, with their size dominance, can create an auto-environment that makes such a molecule independent of environmental influences. However, using the reference (idealized) distributions of *T* and *M*, we can quantify the state of the consensus present between these two idealized environments, leading to the generation of the structure present in these two membrane proteins. Figure 7 presents a comparison of the 3D structures of the proteins in question.

The present analysis indicates that, based on the hydrophobicity distribution characteristics, no barrier exists in the transport of antibiotics through the BamA pore (PDB: 6FSU, 4K3C, and 4N75).

### 3.4. Experimentally Modified Proteins (OprF, CarO1, and CarO2) (PDB: 4RLC, 4RL9I, and 4RLB)

The 4RL9, 4RLB, and 4RLC proteins are discussed together as they were developed by experimental sequence modification for the purpose of understanding the reasons for the observed resistance to antibiotics. In particular, one area of interest in experimental work is the uptake of ornithine and carbapenem [35].

Proteins with ID 4RL9, 4RLB, and 4RLC structures are characterized by the presence of a beta-barrel. The first two feature an additional domain containing the beta sheet and a single helix (Figure 7). As a consequence, in these two proteins (4RL9 and 4RLB), the properties of the beta-barrel itself and its associated domain, which is not present in 4RLC are discussed separately.

The status of the complete molecules is described, with the K values ranging from 1.3 for 4RLB to 0.3 for 4RLC. This wide range reveals the influence of the structural changes deliberately introduced by the experimenters. What is interesting to note is the status of the off-barrel helix, which, in the 4RL9 structure, matches the T distribution, while, in the case of 4RLB, its status clearly differs from the T distribution. The beta-barrel itself shows a significant deviation from the *T* distribution and only the modification at the level of *K* = 1.1 leads to the representation of the *O* distribution. Based on the analysis carried out here, the 4RLC structure, as predicted, showed the highest level of resistance, which is consistent with the experimental results [35]. The out-barrel domain of CarO2 (PDB: 4RLB) diverged dramatically from the standard T distribution, requiring its modification at the level of *K* = 1.8 and leading (similar to the 4N75 protein discussed above) to an *O* distribution similar to the R distribution, which, as was described in the discussion above, suggests that this molecule is independent of the influence of any external environment, thus providing the conditions for folding by itself. This is revealed by a set of profiles (Figure 8) with the structural diversity shown in Figure 9.

### 3.5. Autotransporters (Hbp and EspP) (PDB: 3AEH + 2QOM)

Autotransporters such as Hbp (PDB 3AEH) and EspP (PDB 2QOM) have an N-terminal “passenger” domain, which they can move through their central pore. Although the exact mechanism of autotransport is unclear, the beta barrel domain has been shown to exhibit proteolytic activity. The process is associated with the secretion of the N-terminal “passenger” domain. The beta-barrel exhibits proteolytic activity by digesting the N-terminal domain (“passenger”). The highly specific cut-off site was found between the two adjacent concentrations of asparagine displaying very high levels of conservation. A mutation at this position excludes proteolysis [36]. The location of the “passenger” domains within the beta-barrels of the *E. coli* autotransporters Hbp and EspP are shown in Figure 10.

Figure 11 shows the *T*, *O* and *M* distributions for the two discussed autotransporters: 3AEH and 2QOM.

The profiles show significant deviations from the T distribution (Table 2). These deviations are characteristic of proteins located in the membrane environment, showing hydrophobicity exposure in the surface sections (low expected T values) and decreased hydrophobicity values in the central sections, where a high hydrophobicity is expected (although it is characterized by significant point variations in the level of hydrophobicity in the subsequent residues).

Worth noting is the status of the “passenger” domain, which shows a significantly lower level of hydrophobicity than expected, which is due to the central position of the high hydrophobicity level. The status of the molecule changes as a whole (ranging from high *K* values > 1.0) to levels 0.7 and 0.8 following the cut-off of the N-terminal “passenger” fragment.

It is unlikely that the beta-barrel structure will change significantly in the post-cleaved version. Hence, the correction of the T distribution to *K* = 0.7 and 0.8 can be assumed to be typical for the membrane portions of these proteins.

Based on the example provided by these proteins, the application of the modified FOD model to the FOD-M version, taking into account the influence of the environment on the transmembrane protein structure with the present channel in the central part of the protein yields positive results. These proteins are further examples of the positive application of the FOD-M model as a tool for characterizing these proteins.

### 3.6. A Transmembrane Protein with a Helical Bundle (PAO1) (PDB 5AZO)

The PAO1 protein from *Pseudomonas aeruginosa* is a multidrug efflux pump and is anchored in the membrane by means of a helical bundle. Only part of the complex is available in the PDB, which consists of only two chains out of the six that constitute a complete complex (Figure 12). The status of the yellow and red outside membrane parts can only be judged from the point of view of a single chain. Previous studies [41] show that, in the final complete structure, the beta-barrel composed of extra-membrane parts exhibits a low *RD* for *T**–**O**–**R*, demonstrating significant adaptation to the centric hydrophobicity distribution with considerable discrepancies from the point of view of a single chain.

It can be assumed that this is probably also the case here.

It is difficult to draw any specific conclusions from the 5AZO structure regarding the role of PAO1 in the antibiotic resistance, since the structure is not that of a complete protein complex. Nevertheless, the high *K* values for the entire dimer, the single chain, and the helix system clearly suggest the need for an environmental factor that would stabilize a significantly different hydrophobicity distribution compared to what appears to be common in an aqueous environment (Figure 13).

## 4. Discussion

The analysis presented above aims to correlate the distribution of hydrophobic amino acid residues in bacterial beta-barrel membrane proteins with their function and ability to transport antibiotics. The work revealed that some members of this protein group have a hydrophobicity profile that closely resembles that of globular, soluble proteins. In the case of transmembrane proteins, the exposure of hydrophobic residues to the outside of the molecule is expected with the hydrophobicity deficit in the central part due to the presence of the channel. The opposite distribution to that expected, surprisingly from the point of view of the specific environment in which these proteins are active, explains the bacterial resistance to antibiotics. The structure with a central hydrophobic core with a polar surface does not enable any interaction with other molecules, probably including with antibiotics. For this group of membrane proteins, a potential “antibiotic” should have a structure appropriate to the “target” molecule, whose properties are adjusted to the specificity disclosed here.

The magnitude of the particular *K* parameter reveals the degree of deviation in the distribution with a centric hydrophobic core. The value of this parameter also measures the proportion of the factor that distorts the hydrophobicity distribution from the micelle-like form. In particular, the values of *K* > 1 explain the characteristics of an environment analogous to a “vacuum”, thereby revealing the complete independence of the protein from the influence of a natural water or membrane environment, leading to a form of decomposition close to R decomposition. It can also be interpreted as an environment for intramolecular diffusion in the case of autotransporter proteins. Bacterial proteins are the subject of the analysis in the present work.

Numerous bacterial species live on or inside the human body, including those that facilitate the functioning of the digestive system and are of critical importance for human health [47,48]. The proper functioning of this environment is a vital factor determining the condition of the human body [49]. The aim of the present work was not to solve a specific biological problem. The analysis only showed that the application of the fuzzy oil drop model can serve as a tool for solving the problems of protein–ligand interactions, including the therapeutic purposes in the drug design. The fuzzy oil drop model indicates the presence of a cavity (a local hydrophobicity deficit, as demonstrated in reference [50]). The fuzzy oil drop model was also used in the design of “stoppers” for the propagation of amyloid fibrils [51,52]. When analyzing analogous solutions for naturally occurring and functioning solenoids with a possible tendency to unrestricted complexation, the characterization of “stopper” segments was used to design peptides that could play a similar role for amyloid fibrils.

## 5. Conclusions

The present analysis aimed to demonstrate the validity of the modifications made to the FOD model that take into account the influence (presence) of a hydrophobic environmental factor. The selected proteins are transmembrane proteins and, therefore, have a common environment in the form of a hydrophobic cell membrane. The different values of the *K* parameter (from 0.2 to 1.6) indicate the varying degrees to which the hydrophobic environment shapes their structures, despite the common environment of their activity. In some cases, it is possible to characterize the proteins in question, resulting from the specificity of the distribution of *T*, *O* and *M* in particular. A phenomenon such as a resistance to antibiotics appears to be explained by the status of a given membrane protein, which, contrary to the preliminary assumptions, represents a micelle-like structure (with a hydrophobic core present) and, as such, does not show a tendency to interact with other molecules. A protein that fully represents the hydrophobicity distribution consistent with the Gaussian 3D distribution, apart from local interactions with ions or polar low-molecular compounds, does not exhibit wider interaction possibilities. Such interactions require local inconsistency between the *O* and *T* distributions [44,45]. The analysis of the proteins presented here is a test of the applicability of the FOD model in its modified FOD-M version, revealing the specificity of the proteins from the OmpX group and making it possible to measure the adjustment of the hydrophobicity distribution to the activity environment of these proteins (cell membrane) [53]. This study also showed the usefulness of the FOD and FOD-M models for characterizing the protein structure, especially the relationship with biological activities and/or their possible modifications, including possible drug designs.

## Figures and Tables

**Figure 1 membranes-11-00580-f001:**
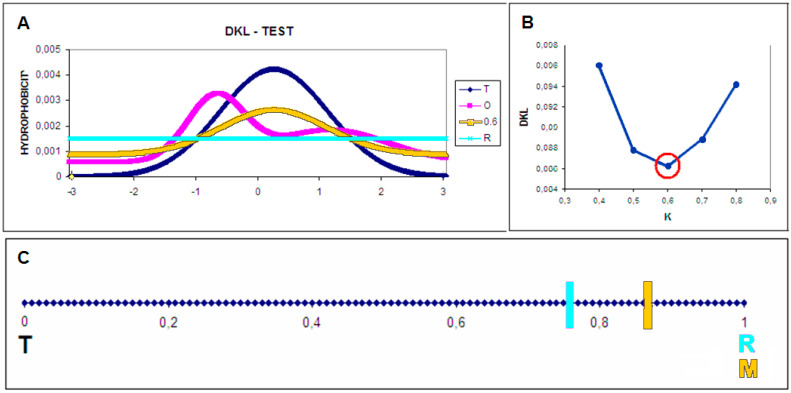
The *O* distribution in relation to the *T*, *M* and *R* distributions (randomly constructed distributions for the 1-dimensional Gauss distribution); (**A**) T distribution (navy), O (pink), R (turquoise), and M (yellow); and (**B**) *D_KL_* values for different values of *K*—the optimal *K* values for the discussed example are marked with red. The *RD* values for the *T**–**O**–**R* and *T**–**O**–**M* relationships are equal to 0.760 and 0.873, respectively. (**C**) The scale of the *RD* values for the reference distributions versus the O distribution. The determined values of *RD* (as given in B) located on the variation axis of the *RD* parameter show significant compatibility with the M distribution compared to the reference distribution *R*.

**Figure 2 membranes-11-00580-f002:**
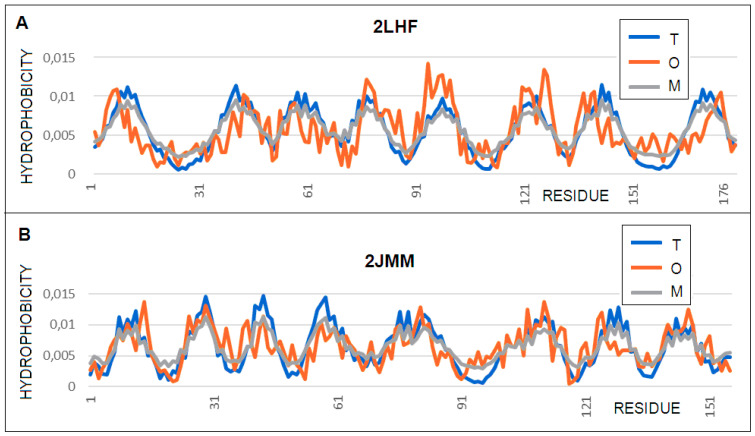
Profiles *T* (blue), *O* (red), and *M* (gray) for: (**A**)—2LHF, with the profile *M* for *K* = 0.2 and (**B)**—2JMM, with the profile *M* for *K* = 0.3.

**Figure 3 membranes-11-00580-f003:**
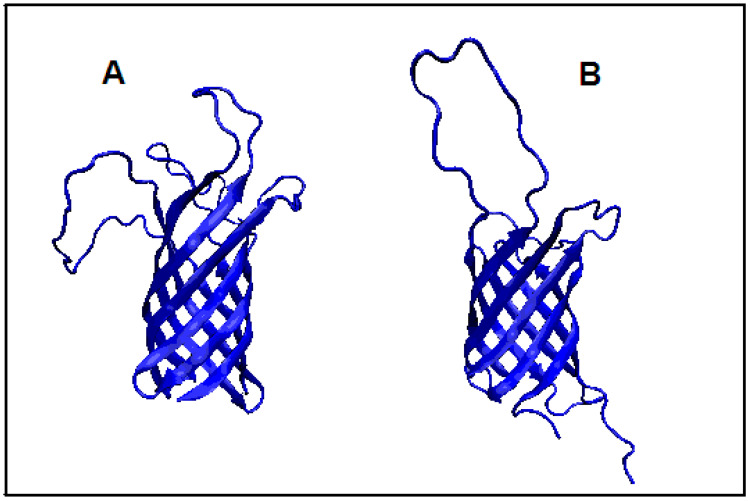
3D structure presentation revealing a different pattern of loops in proteins showing a distribution consistent with the distribution corresponding to the presence of a hydrophobic core. (**A**) 2LHF and (**B**) 2JMM.

**Figure 4 membranes-11-00580-f004:**
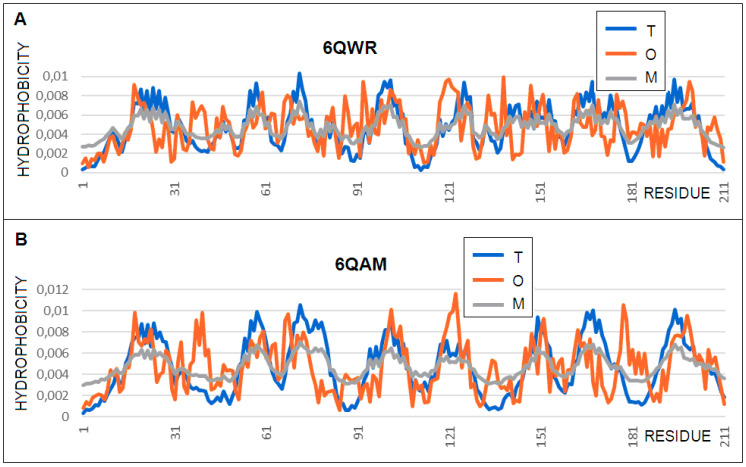
Profiles *T* (blue), *O* (red), and *M* (gray) for: (**A**) the 6QWR–lipid. Profile *M* for *K* = 0.4 and (**B**) 6QAM detergent. Profile *M* for *K* = 0.5.

**Figure 5 membranes-11-00580-f005:**
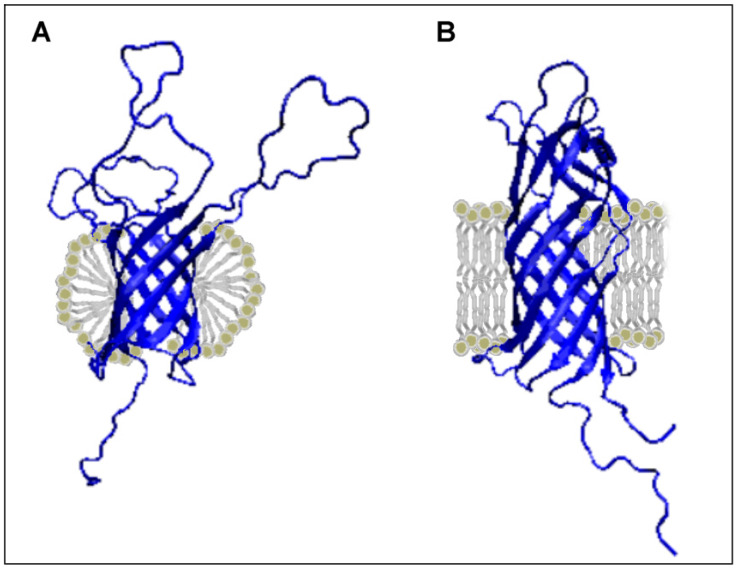
Comparison of the AlkL structures: (**A**) 6QAM detergent environment (gray) and (**B**) 6QWR, lipid environment (gray), according to reference [30].

**Figure 6 membranes-11-00580-f006:**
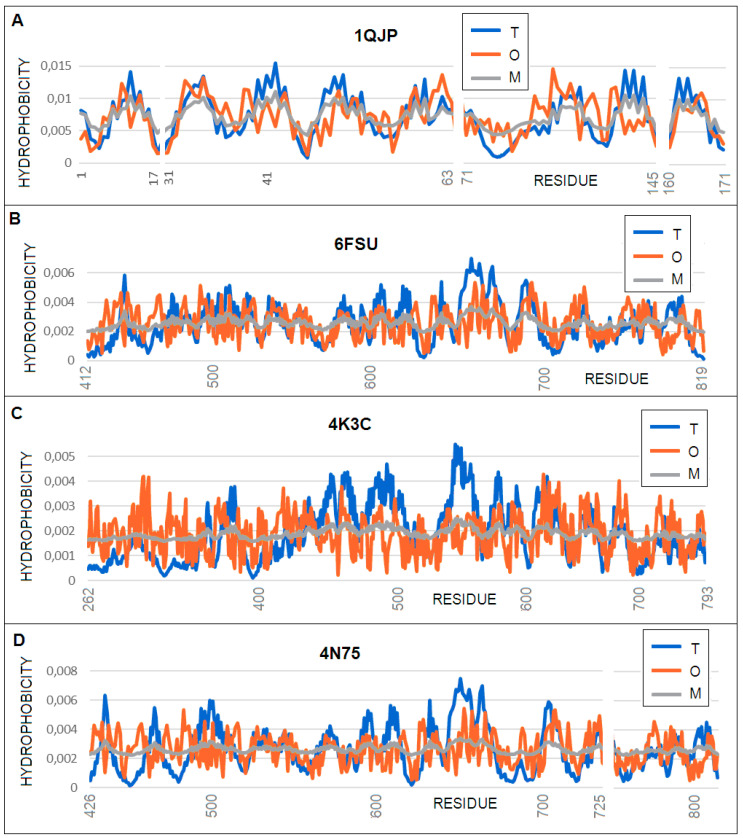
Profiles: *T* (blue), *O* (red), and *M* (gray) for the membrane proteins with a larger tunnel diameter. (**A**) 1QJP, profile *M* for *K* = 0.4; (**B**) 6FSU, profile *M* for *K* = 0.9; (**C**) 4K3C, profile *M* for *K* = 1.2; and (**D**) 4N75, profile *M* for *K* = 1.2.

**Figure 7 membranes-11-00580-f007:**
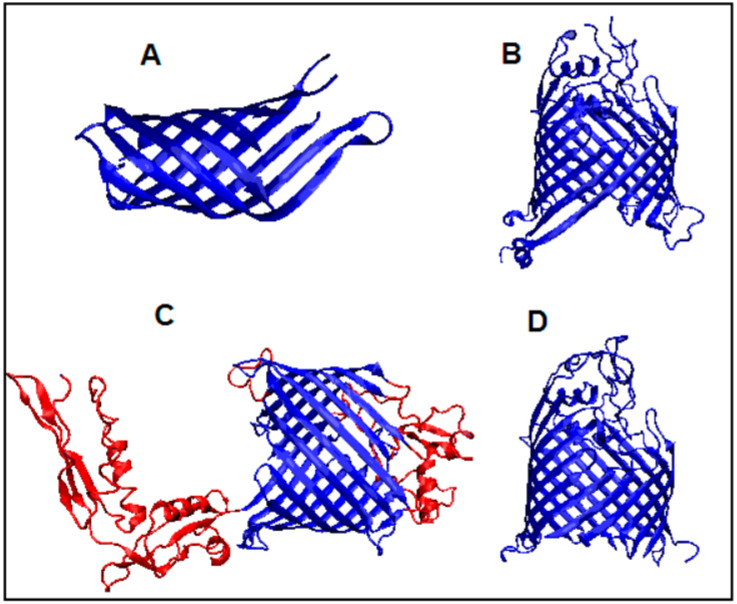
3D presentation of the following proteins: (**A**) 1QJP, (**B**) 6FSU, (**C**) 4K3C with out-of-barrel fragments highlighted in red, and (**D**) 4N75.

**Figure 8 membranes-11-00580-f008:**
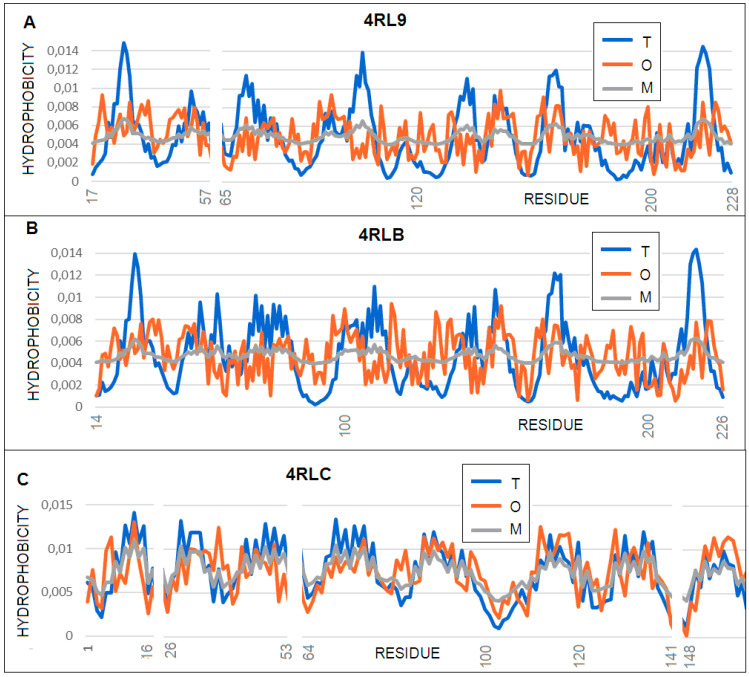
Profiles: *T* (blue), *O* (red), and *M* (gray) for: (**A**) 4RL9, profile *M* for *K* = 1.2; (**B**) 4RLB, profile *M* for *K* = 1.3; and (**C**) 4RLC, profile *M* for *K* = 0.3.

**Figure 9 membranes-11-00580-f009:**
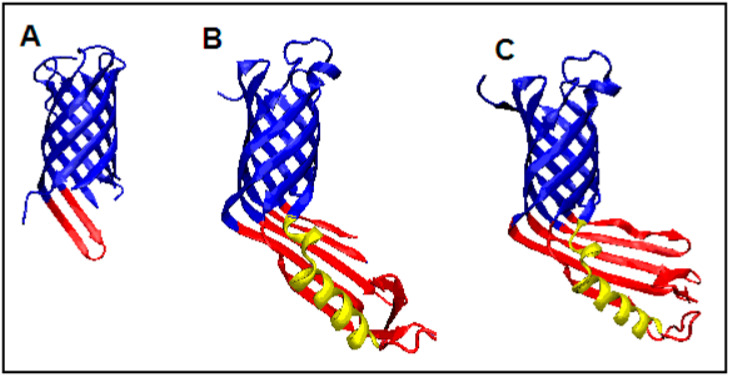
3D presentation of experimentally modified membrane protein structures showing the different statuses of the out-barrel part red. The helix distinguisher is in yellow. (**A**) 2RLC, (**B**) 2RL9, and (**C**) 2RLB.

**Figure 10 membranes-11-00580-f010:**
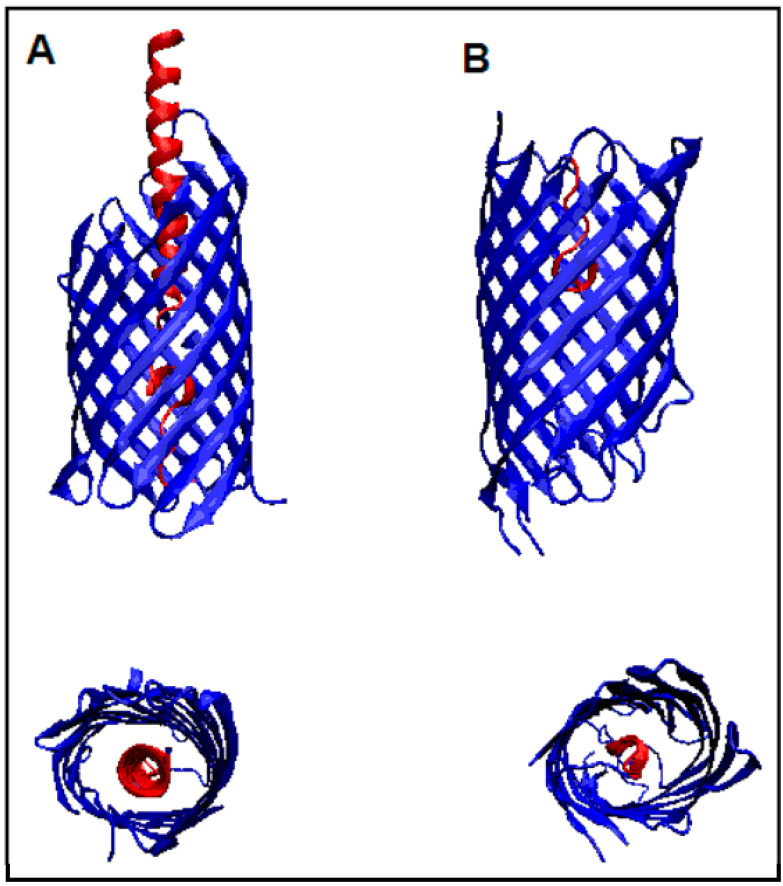
A 3D presentation of the proteins referred to as autotransporters in two different projections. Parts distinguished as red are “passengers”. (A) 3AEH and (B) 2QOM.

**Figure 11 membranes-11-00580-f011:**
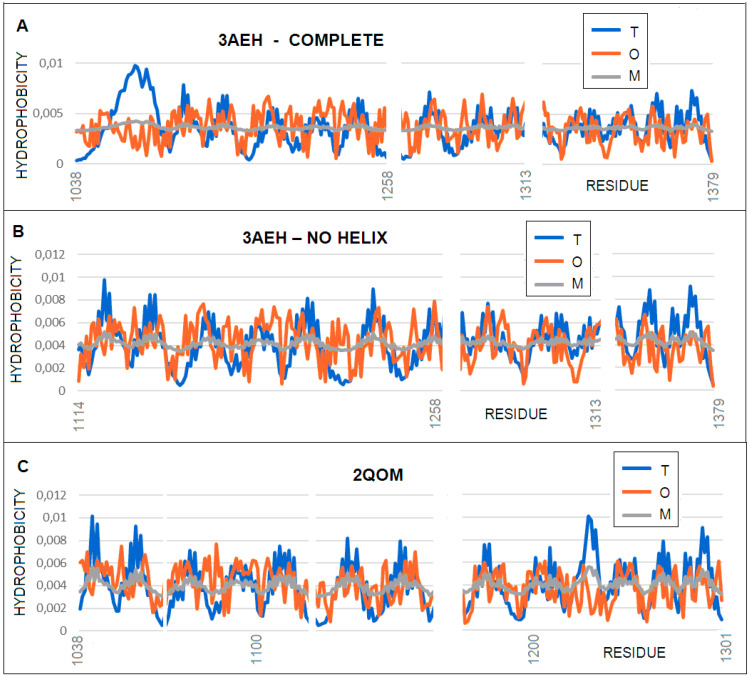
Profiles *T* (blue), *O* (red), and *M* (gray) for: (**A**) 3AEH with a “passenger” present (profile *M* for *K* = 1.3), (**B**) 3AEH lacking an N-terminal helix (“passenger”) (profile *M* for *K* = 0.7), and (**C**) 2QOM without a “passenger” (profile *M* for *K* = 0.8).

**Figure 12 membranes-11-00580-f012:**
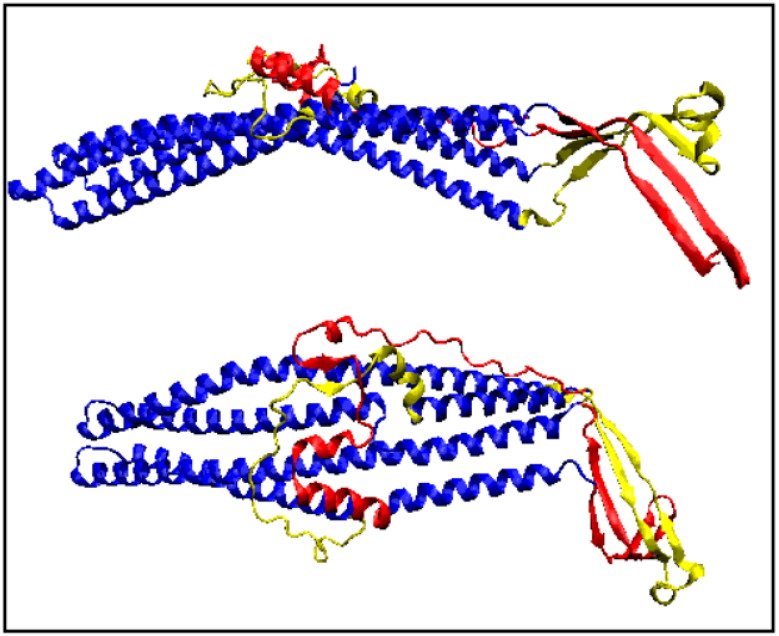
Two different orientations of the dimer structure: a 6-mer section with distinguished extra-membrane parts (red and yellow depending on whether or not they belong to the monomer) and fragments anchored in the membrane (navy blue).

**Figure 13 membranes-11-00580-f013:**
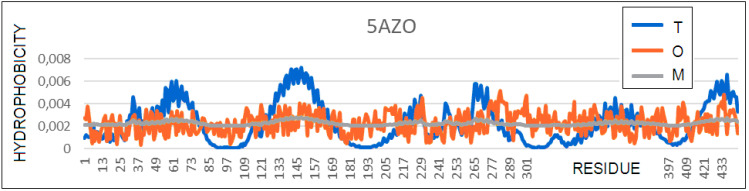
Profiles *T* (blue), *O* (red), and *M* (gray) (*K* = 1.6) of a helical transmembrane protein.

**Table 1 membranes-11-00580-t001:** List of the outer membrane proteins analyzed in the study.

PDB–ID + Chain Length	Protein	Source Organism	Ref.
2LHF–128 aa	Outer membr–oprh	Pseudomonas aeruginosa	[27]
2JMM–165 aa	Outer membr	*Escherichia coli*	[28,29]
6QWR–211 aa	Outer membr–alkl	*Pseudomonas oleovorans*	[30]
	Oil transporter-lipid		
6QAM–211 aa	Outer membr–alkl	*Pseudomonas oleovorans*	[30]
	Oil transporter-detergent		
1QJP–137 aa	Outer membr-ompa	*Escherichia coli*	[31]
4K3C–532 aa	Outer membr-factor bama	*Haemophilus ducreyi*	[32]
6FSU–388 aa	Outer membr-factor bama	*Escherichia coli*	[33]
4N75–379 aa	Outer membr-biogenesis	*Escherichia coli*	[34]
4RLC–135 aa	Outer membr-oprf	*Pseudomonas aeruginosa*	[35]
4RL9–205 aa	Outer memb–carbapenem associated	*Acinetobacter baumannii*	[35]
4RLB-213 aa	Outer memb–carbapenem associated	*Acinetobacter baumannii*	[35]
3AEH–277 aa	Autotransporter-hydrolase	*Escherichia coli*	[36]
2QOM–265 aa	Autotransporter-hydrolase	*Escherichia coli*	[37]
5AZO–444 aa	Efflux pump–oprn	*Pseudomonas aeruginosa*	[38]

**Table 2 membranes-11-00580-t002:** The value of *RD* for the *T*–*O*–*R* relationship and the value of the *K* parameter, which determines the influence of a nonaqueous environment on protein folding, are given. The *RD* values for the *M*–*O*–*T* relationship are also shown, indicating the effect of approximating the M distribution to the *O* distribution at the reference *T* distribution. For the selected proteins, a set of parameters was given for the distinguished structural parts, including, in particular, beta-barrels.

PDB-ID	Protein Characteristics	*RD* *T* *–* *O* *–* *R*	*K*	*RD* *M* *–* *O* *–* *T*	Length (aa)
**HIGH RESISTANCE**					
2LHF	protein H (OprH)	0.472	0.2	0.404	178
β-barrel		0.603	0.4	0.377	74
2JMM	pr. A (OmpA)–modif.	0.472	0.3	0.386	156
β-barrel		0.461	0.2	0.486	84
**DIFFERENT EXTERNAL CONDITIONS**					
6QWR	Oil transport–lipid	0.556	0.4	0.384	211
β-barrel	AlkL	0.537	0.4	0.166	110
Loops		0.599	0.7	0.213	101
6QAM		0.575	0.5	0.365	211
β-barrel	Oil transport–detergent	0.582	0.4	0.415	75
Loops	AlkL	0.676	1.2	0.318	136
**HIGHER DIAMETER BARREL**					
1QJP		0.558	0.4	0.382	137
β-barrel		0.643	0.6	0.349	107
6FSU		0.664	0.9	0.310	388
β-barrel		0.718	0.9	0.281	197
4K3C		0.699	1.2	0.291	532
β-barrel		0.726	1.3	0.273	195
4N75		0.727	1.2	0.261	379
β-barrel		0.743	1.3	0.255	191
**DIFFERENT RESISTANCE**					
4RL9	Small mol. transport	0.745	1.2	0.239	205
β-barrel		0.818	1.1	0.180	76
β-sheet		0.736	1.2	0.262	46
Helix		0.448	0.5	0.448	19
4RLB	Small mol. Transport	0.741	1.3	0.245	213
β-barrel		0.801	1.1	0.196	78
β-sheet		0.806	1.8	0.193	97
Helix		0.651	1.8	0.348	21
4RLC	Small mol. Transport	0.503	0.3	0.409	135
β-barrel		0.503	0.3	0.409	135
**AUTOTRANSPORTER**					
3AEH	autotransporter	0.707	1.3	0.289	277
β-barrel		0.644	0.7	0.336	234
2QOM	autotransporter	0.696	1.4	0.298	265
β-barrel		0.641	0.8	0.346	186
**EFFLUX PUMP**					
5AZO	efflux pump	0.825	1.6	0.169	444
Helices		0.788	1.2	0.206	314
β-sheet		0.837	1.5	0.154	57

**Table 3 membranes-11-00580-t003:** The averaged channel diameters in the discussed proteins.

Structure	Average Diameter (Å)	Protein Name
2HLF	14.2	OprH
6QAM	18.4	AlkL
6QWR	16.3	AlkL
1QJP	15.9	OmpA
6FSU	31.4	BamA
4K3C	35.2	BamA
4N75	32.9	BamA
4RL9	18.3	CarO1
4RLB	12.6	CarO2
4RLC	17.1	OprF
3AEH	24.9	Hbp
2QOM	24.8	EspP

## Data Availability

All data is available on request from the corresponding author. The program enabling the calculation of RD is accessible on request from the CodeOcean platform: https://codeocean.com/capsule/3084411/tree assessed on 28 July 2021. Please contact the corresponding author to gain access to the private program.

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
