# Peer review of "The Functional Significance of Hydrophobic Residue Distribution in Bacterial Beta-Barrel Transmembrane Proteins"

_membranes, 2021, doi:10.3390/membranes11080580_

Round 1

Reviewer 1 Report

This paper analyzes the distribution of hydrophobic and hydrophilic amino acid residues in beta barrel proteins, expressed in the outer membrane of Gram negative bacteria. It attempts to correlate the patterns observed to protein function, with particular reference to the transport of antibiotics. The authors analyse 14 structures from the protein data bank (from 10 distinct proteins) using a modified 3D Gauss function approach based on the fuzzy oil drop model. The method allows the authors to calculate a number of parameters for each structure, which together provide a ‘signature’ for each protein.

Unfortunately, the quality of the writing is very poor and often obscures the scientific analysis and conclusions that the authors are attempting to present. There are frequent mistakes in the use of the English language, but more importantly, the paper is poorly organised. For example, antibiotic resistance is mentioned only at the very end of the abstract, but the majority of the (very short) introduction is focused on it, but it is only mentioned once in the conclusion. Since the introduction is so short, the authors do not fully explain their rationale for choosing the proteins that they study, or how they differ from one another. (There are 2500+ bacterial beta barrel TM protein structures in the pdb – why and how did the authors choose their 14 structures?) The discussion section does not successfully integrate the insights gained from the data presented.

I would recommend that the editor declines this paper and that the authors work to improve the quality of the writing, before resubmitting to the same journal. Ideally a native English speaker should thoroughly proofread the text before resubmission.

Here are some suggested improvements for each of the sections:

Title

- Suggest it is changed to “The functional significance of the distribution of hydrophobic residues in bacterial beta-barrel transmembrane proteins”

Abstract

Antibiotic resistance is only mentioned at the very end of the abstract – should it not be more prominent?

– replace “β-barrel membrane proteins constitute an important group from the point of view of the biological functions, including the transport of certain molecules through the membrane, in particular.” with “β-barrel membrane proteins have several important biological functions, including the transport of water and solutes across the membrane.”

– replace “Compared to proteins that are soluble and active in the aquatic environment, a significant difference is due to the highly hydrophobic environment of the membrane in which the discussed proteins act.” with “They are active in the highly hydrophobic environment of the lipid bilayer, unlike soluble proteins, which function in a more polar, aqueous environment.”

– replace “Globular soluble proteins represent a centric concentration of hydrophobicity (hydrophobic core) with a polar surface providing favorable contact with the polar environment of the water.” with “Globular soluble proteins typically have a hydrophobic core and a polar surface which interacts favorably with water.”

Introduction

The introduction is very short and there is too much emphasis on antibiotic resistance, which is only a secondary focus on the paper. The authors should briefly review existing computational methods for analysing bacterial outer membrane proteins and explain why there is a need for a new methodology to be developed.

– lines 30-32 – replace “The object of the analysis are transmembrane proteins present in outer membrane of the bacteria. The knowledge of the structures and properties of these proteins is important for therapeutic techniques with the use of antibiotics [1-3].” with “β-barrel transmembrane proteins are expressed in the outer membrane of bacterial cells. Since mutations in these proteins have been implicated in conferring antibiotic resistance, knowledge of their structures and functions important for the development of clinical therapies.”

– lines 32-35 – replace “Bacteria have developed mechanisms of resistance to the antibiotics [4-6]. This phenomenon is particularly dangerous due to the so-called hospital infections [7-11]. Rapid appearance of resistance to a specific antibiotic is defined as the misuse effect that eliminates the antibiotic from therapeutic processes [12-14]. This drives the development of new solutions [15-17].” with “The prevalence of drug resistance, especially in bacteria strains implicated in hospital-acquired infections, have important implications for modern medicine. New solutions to overcome resistance, which commonly arises from the misuse and over-prescription of antibiotics, requires the development of new solutions.”

– lines 39-41 – replace “Independently developed research within the so-called systems biology comes to the rescue, which by looking for the mechanisms of action of a living organism, including protein-protein interactions and the network of such connections, can be effectively used in research on new antibiotics [21, 22].” with “Systems biology approaches that integrate knowledge from structural and functional studies of transport proteins and related signalling molecules must be used in the development of new antibiotics.”

– lines 43-46 – replace “The object of these analyzes is the very phenomenon of resistance that bacteria develop to the successively developed antibiotics [23-30]. The analysis of this phenomenon at the molecular level focuses on the stage of the mechanism of penetration of the designed antibiotic into the bacterial canal.” with “In order to enter the cell of a Gram-negative bacterium, an antibiotic must cross through the outer membrane. A molecular understanding of how drug molecules pass through the channels in this membrane is essential in the development of new therapeutic compounds.”

Materials and methods

– much of subsection 2.2 is confusingly written, especially the explanation of the T-O-R and M-O-T relationships.

Table 1

– should include the names of their proteins as well as their functions (and molecular weights)

Table 2

– make it clear that the 6QWR and 6QAM structures are of the same AlkL protein

– how does the barrel diameter of 1QJP, 6FSU, 4K3C, 4N75 compare to other beta barrels – give a quantitative measure

– check the spelling of “efflux pomp”

Figs 2, 4, 6, 8, 9

– should have a white background

– should indicate the membrane topology

– figure legends should explain color scheme

Discussion

– section should compare T-O-M profiles of the different proteins studied more comprehensively

Author Response

REVIEWER I

Comments and Suggestions for Authors

Many thanks for comments. We did our best to follow your suggestions.

This paper analyzes the distribution of hydrophobic and hydrophilic amino acid residues in beta barrel proteins, expressed in the outer membrane of Gram negative bacteria. It attempts to correlate the patterns observed to protein function, with particular reference to the transport of antibiotics. The authors analyse 14 structures from the protein data bank (from 10 distinct proteins) using a modified 3D Gauss function approach based on the fuzzy oil drop model. The method allows the authors to calculate a number of parameters for each structure, which together provide a ‘signature’ for each protein.

 Unfortunately, the quality of the writing is very poor and often obscures the scientific analysis and conclusions that the authors are attempting to present. There are frequent mistakes in the use of the English language, but more importantly, the paper is poorly organised. For example, antibiotic resistance is mentioned only at the very end of the abstract, but the majority of the (very short) introduction is focused on it, but it is only mentioned once in the conclusion. Since the introduction is so short, the authors do not fully explain their rationale for choosing the proteins that they study, or how they differ from one another. (There are 2500+ bacterial beta barrel TM protein structures in the pdb – why and how did the authors choose their 14 structures?) The discussion section does not successfully integrate the insights gained from the data presented.

 I would recommend that the editor declines this paper and that the authors work to improve the quality of the writing, before resubmitting to the same journal. Ideally a native English speaker should thoroughly proofread the text before resubmission.

 Here are some suggested improvements for each of the sections:

 Title

  • Suggest it is changed to “The functional significance of the distribution of hydrophobic residues in bacterial beta-barrel transmembrane proteins”

The title has been changed according to the suggestion

 Abstract

Antibiotic resistance is only mentioned at the very end of the abstract – should it not be more prominent?

  • replace “β-barrel membrane proteins constitute an important group from the point of view of the biological functions, including the transport of certain molecules through the membrane, in particular.” with “β-barrel membrane proteins have several important biological functions, including the transport of water and solutes across the membrane.”

It has been changed

  • replace “Compared to proteins that are soluble and active in the aquatic environment, a significant difference is due to the highly hydrophobic environment of the membrane in which the discussed proteins act.” with “They are active in the highly hydrophobic environment of the lipid bilayer, unlike soluble proteins, which function in a more polar, aqueous environment.”

It has been changed

  • replace “Globular soluble proteins represent a centric concentration of hydrophobicity (hydrophobic core) with a polar surface providing favorable contact with the polar environment of the water.” with “Globular soluble proteins typically have a hydrophobic core and a polar surface which interacts favorably with water.”

It has been changed

 Introduction

The introduction is very short and there is too much emphasis on antibiotic resistance, which is only a secondary focus on the paper. The authors should briefly review existing computational methods for analysing bacterial outer membrane proteins and explain why there is a need for a new methodology to be developed.

  • lines 30-32 – replace “The object of the analysis are transmembrane proteins present in outer membrane of the bacteria. The knowledge of the structures and properties of these proteins is important for therapeutic techniques with the use of antibiotics [1-3].” with “β-barrel transmembrane proteins are expressed in the outer membrane of bacterial cells. Since mutations in these proteins have been implicated in conferring antibiotic resistance, knowledge of their structures and functions important for the development of clinical therapies.”

It has been changed

  • lines 32-35 – replace “Bacteria have developed mechanisms of resistance to the antibiotics [4-6]. This phenomenon is particularly dangerous due to the so-called hospital infections [7-11]. Rapid appearance of resistance to a specific antibiotic is defined as the misuse effect that eliminates the antibiotic from therapeutic processes [12-14]. This drives the development of new solutions [15-17].” with “The prevalence of drug resistance, especially in bacteria strains implicated in hospital-acquired infections, have important implications for modern medicine. New solutions to overcome resistance, which commonly arises from the misuse and over-prescription of antibiotics, requires the development of new solutions.”

It has been changed

  • lines 39-41 – replace “Independently developed research within the so-called systems biology comes to the rescue, which by looking for the mechanisms of action of a living organism, including protein-protein interactions and the network of such connections, can be effectively used in research on new antibiotics [21, 22].” with “Systems biology approaches that integrate knowledge from structural and functional studies of transport proteins and related signaling molecules must be used in the development of new antibiotics.”

It has been changed

  • lines 43-46 – replace “The object of these analyzes is the very phenomenon of resistance that bacteria develop to the successively developed antibiotics [23-30]. The analysis of this phenomenon at the molecular level focuses on the stage of the mechanism of penetration of the designed antibiotic into the bacterial canal.” with “In order to enter the cell of a Gram-negative bacterium, an antibiotic must cross through the outer membrane. A molecular understanding of how drug molecules pass through the channels in this membrane is essential in the development of new therapeutic compounds.”

 It has been changed

 Materials and methods

  • much of subsection 2.2 is confusingly written, especially the explanation of the T-O-R and M-O-T relationships.

The description of the analysis for the M-O-T and T-O-R relations has been added in Materials and Methods

 Table 1

  • should include the names of their proteins as well as their functions (and molecular weights)
  •  
  • It has been completed

 Table 2

  • make it clear that the 6QWR and 6QAM structures are of the same AlkL protein

It has been completed

  • how does the barrel diameter of 1QJP, 6FSU, 4K3C, 4N75 compare to other beta barrels – give a quantitative measure

The table with appropriate values has been added in the sub-chapter : “The comparative analysis”

– check the spelling of “efflux pomp” – it has been corrected

 Figs 2, 4, 6, 8, 9

– should have a white background – the white background is given

– should indicate the membrane topology – it has been added

– figure legends should explain color scheme – it has been corrected

 Discussion

– section should compare T-O-M profiles of the different proteins studied more comprehensively

The newly added sub-chapter explains in details the idea of M distribution. The Figure 1 is expected to clarify this problem

Reviewer 2 Report

The manuscript addresses an important issue regarding the insertion and modification of transmembrane proteins dependent on the B-barrel structure. In addition, it seeks to validate the findings of applying analytical software (FOD) to infer microbial resistance to antibiotics using several examples of proteins. Without a doubt, the work is significant; however, the manuscript is not well presented and leaves something to be desired regarding the presentation and fluidity of the subject.

Major points

The abstract is confused and does not inform what the objective of the work is. In the same way, the presentation of the results and the discussion could be more exploratory, confronting, and discussing the findings with the Alpha-Helix-dependent transmembrane models.

Other Minor points

1) L13- change aquatic x aqueous

2) L18-"Membrane protein......" -I disagree with the statement since there are other models of transmembrane proteins

3) The abstract does not inform the objective of the work. However, in the first sentence of the introduction, it talks about the purpose but does not inform what the job is intended for

4) The introduction must rewrite- The subject is presented with watertight sentences, many of which are meaningless with the following sentence.

5) L30, L43, and L49-informs different objectives; after all, what is the aim of the work?

6) L63-“This solution is surprising…”-Please modify this sentence as the results are not so surprising since the work only addresses a select group of B-barrel-dependent proteins, and the sense of the face is much broader. Many transmembrane proteins do not conform to this pattern like class 1 and alpha-helix dependent proteins.

7) Fig 4, L250, L251- The letters are in lowercase, and the text in uppercase in the figure.

Author Response

REVIEWER II

Comments and Suggestions for Authors

Many thanks for suggestions. We did our best to follow your comments.

The manuscript addresses an important issue regarding the insertion and modification of transmembrane proteins dependent on the B-barrel structure. In addition, it seeks to validate the findings of applying analytical software (FOD) to infer microbial resistance to antibiotics using several examples of proteins. Without a doubt, the work is significant; however, the manuscript is not well presented and leaves something to be desired regarding the presentation and fluidity of the subject.

 Major points

The abstract is confused and does not inform what the objective of the work is. In the same way, the presentation of the results and the discussion could be more exploratory, confronting, and discussing the findings with the Alpha-Helix-dependent transmembrane models.

The analysis of another transmembrane protein - rhodopsin - was attached to the text of the work in order to explain the possibility of using the fuzzy oil drop model, especially in its modified form, taking into account the amphipathic environment of the cell membrane.

Other Minor points

1) L13- change aquatic x aqueous – it has been corrected

2) L18-"Membrane protein......" -I disagree with the statement since there are other models of transmembrane proteins – other positions in References have been added 

3) The abstract does not inform the objective of the work. However, in the first sentence of the introduction, it talks about the purpose but does not inform what the job is intended for

4) The introduction must rewrite- The subject is presented with watertight sentences, many of which are meaningless with the following sentence.

INTRODUCTION – it has been changed

5) L30, L43, and L49-informs different objectives; after all, what is the aim of the work?

The clearly stated aim of the work has been added at the end of Introduction section.

6) L63-“This solution is surprising…”-Please modify this sentence as the results are not so surprising since the work only addresses a select group of B-barrel-dependent proteins, and the sense of the face is much broader. Many transmembrane proteins do not conform to this pattern like class 1 and alpha-helix dependent proteins.

According to the assumption that the hydrophobic (amphipathic) environment of the membrane expects mainly hydrophobic contact with the protein, the presence of a structure consistent with the 3D-Gauss distribution does not comply with this assumption. In this sense, the word "surprise" was used. This word was changed to "not as expected"

7) Fig 4, L250, L251- The letters are in lowercase, and the text in uppercase in the figure.

It has been corrected

Reviewer 3 Report

In the present experimental study, Irena Roterman and coworkers focused on the structures of bacterial proteins from the OmpX group, analyzing their adaptation to the hydrophobic environment of the membrane and preparation for the transport of various molecules. The analysis is based on the use of the fuzzy oil drop (FOD) model and in particular, the modified version of this model, the FOD-M model, where M expresses the membrane environment. With the help of this analysis, it is possible to explain the specific features of the proteins in question, such as their differentiated resistance to antibiotics of some forms and their specificity in relation to the transported molecule, such as oil transport through a membrane. The authors concluded that the analysis of the proteins presented in the current paper is a test for the applicability of the FOD model in its modified FOD-M version revealing the specificity of proteins from the OmpX group, allowing the identification of the degree of adjustment of the hydrophobicity distribution to the activity environment of these proteins (cell membrane). Overall, I think that the paper is well organized, well-structured, timely and it could be of interest to the readers of Membranes and researchers, in general.

I raise a series of minor points to address carefully for improve, in my opinion, the overall quality of paper.

1) It is possible develop the present analysis in order to study the membrane of gut microbiote? Please add, eventually, further details and please insert appropriate references.

2) I have a little curiosity. The FOD model allow to define the degree of adjustment of the hydrophobicity distribution to the activity environment of these proteins. It is possible assess the resistance only for antibiotics? In other words, it is possible for example to test different drug class utilized in clinical practice (i.e. antioxidants/anti-inflammatory drugs, nutraceuticals etc.? Please discuss this topic. I feel that your answer could have interesting perspectives from a translational point of view in humans.

Author Response

REVIEWER III

Comments and Suggestions for Authors

Many thanks for comments. We did our best to follow your suggestions.

In the present experimental study, Irena Roterman and coworkers focused on the structures of bacterial proteins from the OmpX group, analyzing their adaptation to the hydrophobic environment of the membrane and preparation for the transport of various molecules. The analysis is based on the use of the fuzzy oil drop (FOD) model and in particular, the modified version of this model, the FOD-M model, where M expresses the membrane environment. With the help of this analysis, it is possible to explain the specific features of the proteins in question, such as their differentiated resistance to antibiotics of some forms and their specificity in relation to the transported molecule, such as oil transport through a membrane. The authors concluded that the analysis of the proteins presented in the current paper is a test for the applicability of the FOD model in its modified FOD-M version revealing the specificity of proteins from the OmpX group, allowing the identification of the degree of adjustment of the hydrophobicity distribution to the activity environment of these proteins (cell membrane). Overall, I think that the paper is well organized, well-structured, timely and it could be of interest to the readers of Membranes and researchers, in general.

I raise a series of minor points to address carefully for improve, in my opinion, the overall quality of paper.

1) It is possible develop the present analysis in order to study the membrane of gut microbiote? Please add, eventually, further details and please insert appropriate references.

The aim of the publication is to demonstrate the applicability of the fuzzy oil drop model to design (predict) a potential interaction of a given protein with a natural ligand or a designed candidate drug.

This comment has been added to the work.

2) I have a little curiosity. The FOD model allow to define the degree of adjustment of the hydrophobicity distribution to the activity environment of these proteins. It is possible assess the resistance only for antibiotics? In other words, it is possible for example to test different drug class utilized in clinical practice (i.e. antioxidants/anti-inflammatory drugs, nutraceuticals etc.? Please discuss this topic. I feel that your answer could have interesting perspectives from a translational point of view in humans.

The application of the fuzzy oil drop model is not limited to predicting the possibility of interaction with any molecule including antibiotics in particular. The discussed model was used to design peptides stopping the development of amyloid fibrils. This proposal is based on the analysis of the structures of solenoids, which have specific "stop" fragments in their structure. This issue is discussed in detail in “Anti-amyloid drug design “, Banach M, Konieczny L, Roterman I, in the book “From globula proteins to amyloids”,  Elsevier 2020 pp 215-232 and Roterman I, Banach M, Konieczny L. “Towards the design of anti-amyloid short peptide helices”. Bioinformation. 2018: 14 (1), pp. 1-7.

This fragment has been added to the publication.

Round 2

Reviewer 1 Report

The authors have worked to improve the quality of the written English in this paper, however it would benefit from further improvement. I have made some further suggested changes below, but in my view it is imperative that the text is edited by a native English speaker before publication to ensure that the paper can be fully understood by its readers.

line 19 - replace “and in the case of the ion channel in the central part concentrating the polar residues” with "and, in the case of ion channels, the polar residues are concentrated in the central part"

line 53 - replace "These are proteins located in the outer membrane of bacteria that communicate with the outside world with the periplasmic space of the bacteria" to "These are proteins located in the outer membrane of bacteria that connect the periplasm to the outside world"

line 75 - replace "The analysis of these proteins was carried out on the basis of the fuzzy oil drop model with modifications (FOD-M), for which the hydrophobicity distribution is the criterion" with "The analysis of these proteins was carried out using a modified form of the fuzzy oil drop model (FOD-M), for which the hydrophobicity distribution is the key criterion"

Table 1 - Latin names of source organisms should be in italics

line 91 - replace "Both the fuzzy oil drop model (FOD) and its modification of FOD-M taking into account the presence and influence of non-polar factors on shaping the structure of proteins have been presented in numerous papers" with "Both the fuzzy oil drop model (FOD) and its modified form (FOD-M), which takes into account the influence of the non-polar environment on membrane protein structure, have been presented in numerous papers"

line 114 - check the spelling of “distributions”

line 134 - replace "Here, the characteristics of the environment expect exposure of hydrophobic residues to the surface" with "Here, the non-polar character of the lipid environment is expected to promote the exposure of hydrophobic residues on the protein surface."

line 135 - replace "Additionally, if the membrane protein acts as a channel, and an ion channel in particular, the distribution of hydrophobicity requires a polar channel to perform its function, which is generally centered." with ""Additionally, if the membrane protein acts as a channel, and especially if it acts as an ion channel, polar residues are expected to point inwards to stabilise a water-filled internal pore."

line 226 - replace "Exposure of hydrophobic residues in beta-barrel segments in contact with the membrane is rather expected (Figure 3)" with "Exposure of hydrophobic residues in beta-barrel segments in contact with the membrane is expected instead (Figure 3)"

line 242 - replace "Protein H (OprH) (PDB ID 2LHF, pseudomonas aeruginosa) is regarded as an impermeability factor causing notorious antibiotic resistance"  with "Mutations in Protein H (OprH; PDB ID 2LHF) are implicated in causing antibiotic resistance in Pseudomonas aeruginosa"

line 264 - replace "The fuzzy oil drop model does not explain the reasons why the hydrophobicity distribution in these analyzed proteins contradicts the expectations related to the structure and specificity of the transmembrane barrel" with "The fuzzy oil drop model does not explain why the distribution of hydrophobic residues does not correlate with the structure of the transmembrane beta barrel."

line 278 - replace "The protein AlkL increasing permeability of the outer membrane of bacteria for hydrophobic molecules in two structural forms differentiated by environment is present in this analysis to quantitatively estimate the influence of external force field in form of lipid (6QWR) and detergent (6QAM)" with ""The protein AlkL increases the permeability of the Pseudomonas oleovorans outer membrane to hydrophobic molecules. Two structures, one solved in the presence of lipid (6QWR) and one in the presence of detergent (6QAM) are analysed in this study to determine the importance of the composition of the hydrophobic environment."

line 323 - replace "The summary includes a protein with a diameter comparable to the previously discussed" with "The summary includes a protein with a diameter comparable to those previously discussed"

line 345 - replace "There is an increasing values of O in the sections exposed to the environment" with "There is an increase in values of O in the sections of the protein exposed to the environment."

line 346 - replace "It ca be visible the widening of the range of sections showing a deficit in the positions of the expected maximums" with "A wide range of sections show a large deficit between the T and O maxima."

line 410 - replace "The presence of the "passenger" domain is visualized in Figure 10" with "The location of the "passenger" domains within the beta barrels of the E. coli autotransporters Hbp and EspP, are shown in Figure 10."

line 480 - replace "The analysis of the proteins discussed herein is an overview of the bacterial outer membrane proteins" with "The analysis presented here aims to correlate the distribution of hydrophobic amino acid residues in bacterial beta barrel membrane proteins with their function and ability to transport antibiotics."

line 481 - replace "The analysis provides some surprises regarding the status of membrane proteins showing a hydrophobicity distribution consistent with that observed in globular soluble proteins" with "The work has revealed that some members of this group of proteins have a hydrophobicity profile that closely resembles that of globular, soluble proteins."

line 500 - replace "Numerous bacterial resources in the human body are the Gut microbiota environment. The proper functioning of this environment is of critical importance for the condition of the human body" with "Numerous bacterial species live on or inside the human body, including those which facilitate the functioning of the digestive system, and are of critical importance for human health."

Author Response

REVIEWER I

We followed the suggestions given below. Many thanks for your work to improve the quality of our paper. We also introduced corrections given by native speaker.

Many thanks for your comments.

Sincerely yours;

Irena Roterman

Comments and Suggestions for Authors

The authors have worked to improve the quality of the written English in this paper, however it would benefit from further improvement. I have made some further suggested changes below, but in my view it is imperative that the text is edited by a native English speaker before publication to ensure that the paper can be fully understood by its readers.

line 19 - replace “and in the case of the ion channel in the central part concentrating the polar residues” with "and, in the case of ion channels, the polar residues are concentrated in the central part"

line 53 - replace "These are proteins located in the outer membrane of bacteria that communicate with the outside world with the periplasmic space of the bacteria" to "These are proteins located in the outer membrane of bacteria that connect the periplasm to the outside world"

line 75 - replace "The analysis of these proteins was carried out on the basis of the fuzzy oil drop model with modifications (FOD-M), for which the hydrophobicity distribution is the criterion" with "The analysis of these proteins was carried out using a modified form of the fuzzy oil drop model (FOD-M), for which the hydrophobicity distribution is the key criterion"

Table 1 - Latin names of source organisms should be in italics

line 91 - replace "Both the fuzzy oil drop model (FOD) and its modification of FOD-M taking into account the presence and influence of non-polar factors on shaping the structure of proteins have been presented in numerous papers" with "Both the fuzzy oil drop model (FOD) and its modified form (FOD-M), which takes into account the influence of the non-polar environment on membrane protein structure, have been presented in numerous papers"

line 114 - check the spelling of “distributions”

line 134 - replace "Here, the characteristics of the environment expect exposure of hydrophobic residues to the surface" with "Here, the non-polar character of the lipid environment is expected to promote the exposure of hydrophobic residues on the protein surface."

line 135 - replace "Additionally, if the membrane protein acts as a channel, and an ion channel in particular, the distribution of hydrophobicity requires a polar channel to perform its function, which is generally centered." with ""Additionally, if the membrane protein acts as a channel, and especially if it acts as an ion channel, polar residues are expected to point inwards to stabilise a water-filled internal pore."

line 226 - replace "Exposure of hydrophobic residues in beta-barrel segments in contact with the membrane is rather expected (Figure 3)" with "Exposure of hydrophobic residues in beta-barrel segments in contact with the membrane is expected instead (Figure 3)"

line 242 - replace "Protein H (OprH) (PDB ID 2LHF, pseudomonas aeruginosa) is regarded as an impermeability factor causing notorious antibiotic resistance"  with "Mutations in Protein H (OprH; PDB ID 2LHF) are implicated in causing antibiotic resistance in Pseudomonas aeruginosa"

line 264 - replace "The fuzzy oil drop model does not explain the reasons why the hydrophobicity distribution in these analyzed proteins contradicts the expectations related to the structure and specificity of the transmembrane barrel" with "The fuzzy oil drop model does not explain why the distribution of hydrophobic residues does not correlate with the structure of the transmembrane beta barrel."

line 278 - replace "The protein AlkL increasing permeability of the outer membrane of bacteria for hydrophobic molecules in two structural forms differentiated by environment is present in this analysis to quantitatively estimate the influence of external force field in form of lipid (6QWR) and detergent (6QAM)" with ""The protein AlkL increases the permeability of the Pseudomonas oleovorans outer membrane to hydrophobic molecules. Two structures, one solved in the presence of lipid (6QWR) and one in the presence of detergent (6QAM) are analysed in this study to determine the importance of the composition of the hydrophobic environment."

line 323 - replace "The summary includes a protein with a diameter comparable to the previously discussed" with "The summary includes a protein with a diameter comparable to those previously discussed"

line 345 - replace "There is an increasing values of O in the sections exposed to the environment" with "There is an increase in values of O in the sections of the protein exposed to the environment."

line 346 - replace "It ca be visible the widening of the range of sections showing a deficit in the positions of the expected maximums" with "A wide range of sections show a large deficit between the T and O maxima."

line 410 - replace "The presence of the "passenger" domain is visualized in Figure 10" with "The location of the "passenger" domains within the beta barrels of the E. coli autotransporters Hbp and EspP, are shown in Figure 10."

line 480 - replace "The analysis of the proteins discussed herein is an overview of the bacterial outer membrane proteins" with "The analysis presented here aims to correlate the distribution of hydrophobic amino acid residues in bacterial beta barrel membrane proteins with their function and ability to transport antibiotics."

line 481 - replace "The analysis provides some surprises regarding the status of membrane proteins showing a hydrophobicity distribution consistent with that observed in globular soluble proteins" with "The work has revealed that some members of this group of proteins have a hydrophobicity profile that closely resembles that of globular, soluble proteins."

line 500 - replace "Numerous bacterial resources in the human body are the Gut microbiota environment. The proper functioning of this environment is of critical importance for the condition of the human body" with "Numerous bacterial species live on or inside the human body, including those which facilitate the functioning of the digestive system, and are of critical importance for human health."

Reviewer 2 Report

The main suggestions were accepted by the authors, and the manuscript received a good improvement, so we consider that the manuscript can now be accepted for publication. However, some minor errors and spelling should be corrected.

Author Response

REVIEWER II

Many thanks for your suggestion. We engaged the native speaker to correct our paper.

Sincerely yours;

Irena Roterman

Comments and Suggestions for Authors

The main suggestions were accepted by the authors, and the manuscript received a good improvement, so we consider that the manuscript can now be accepted for publication. However, some minor errors and spelling should be corrected.

Round 3

Reviewer 1 Report

The authors have asked a native English speaker to check through their manuscript and to improve the quality of the writing. Unfortunately only very minor changes (mostly spelling) have been made and no serious attempt has been made to improve the grammar and the clarity of large sections. Here are some further suggested amendments.

I would urge the authors to ask a second native English speaker to work on the manuscript to improve the quality of the writing. At present the manuscript does not do justice to the scientific work being presented. 

IMPORTANT - a PDB code (e.g. 2LHF or 2JMM) indicates a STRUCTURE (from X-ray crystallography or NMR) and is not the same as the name of a protein (e.g. OprH or OmpA)

IMPORTANT - please make sure that all Latin organism names are italicized

- line 17 - replace "Membrane proteins represent the opposite system, exposing hydrophobic residues on the surface and, in the case of ion channels, the polar residues are concentrated in the central part" with ""Membrane proteins represent the opposite system, exposing hydrophobic residues on the surface and, in the case of ion channels, the polar residues face inwards towards a central pore."

- line 19 - replace "Their hydrophobicity distribution can be expressed using the complementary function to Gaussian (3DG) in the form 1-3DG" with "The distribution of hydrophobic residues in membrane proteins can be characterised using 1-3DG, a modified 3D Gauss function."

- line 23 - replace "The identified differentiation translates into the specificity of the biological activity of these proteins, such as the type of molecule being transported and the resistance to antibiotics" with "The biological activity and substrate-specificity of these proteins, is determined by the distribution of polar and non-polar amino acids"

- line 25 - replace "Comparative results help to identify the degree of resistance to antibiotics based on the possibility of interaction of a given protein with molecules resulting from the structure of the hydrophobic core" with "The analysis allows us to compare the ways in which the different proteins interact with antibiotics and helps us to understand their relative importance for development of resistance mechanisms."

- line 27 - I am not sure what is meant here. It might be better to replace "Centrally located hydrophobic core introducing stabilization within the molecule is treated as a factor that prevents additional interactions with other molecules" with "We show that beta barrel membrane proteins with a hydrophobic core interact less strongly with the molecules that they transport." But I do not really understand what the authors are trying to say in this sentence.

- line 36 - replace "knowledge of their structures and functions appears important" with "knowledge of their structures and functions is important".

- line 39 - replace "New solutions to overcome resistance, which commonly arises from the misuse and over-prescription of antibiotics, require the development of new solutions" with "New solutions to overcome resistance, which commonly arises from the misuse and over-prescription of antibiotics, are required."

- line 50 - replace "The analysis of this phenomenon at the molecular level focuses on the stage of the mechanism of penetration of the designed antibiotic into the bacterial canal" with "The analysis of this phenomenon at the molecular level focuses on the mechanism of penetration of the designed antibiotic into transmembrane channels."

- line 53 - replace "These are proteins located in the outer membrane of bacteria that 53 connect the periplasm to the outside world" with "These proteins are located in the outer membrane of bacteria and connect the periplasm to the outside world."

- line 69 - check spelling of "heliacal"

- line 96 - replace "The FOD model assumes the possibility of describing the hydrophobicity distribution in globular proteins using a 3D Gaussian function" with "The FOD model uses a 3D Gaussian function to describe the distribution of hydrophobic amino acid residues in globular proteins."

- line 114 - correct "where P the analyzed <<< missing words after “P”"

- line 199 - replace "The summary of the obtained results for the description of the status of the discussed membrane proteins is presented in Table 2" with "Table 2 summarizes the results of the FOD-M analysis for members of the OmpX beta barrel membrane protein family."

- line 211 - replace "The K values determine the degree of participation of a factor other than polar in shaping the structure" with "The K values indicate the relative importance of polar and non-polar residues in determining the protein structure."

- line 211 and 220 - duplication?

- line 225 - replace "3.1. Proteins resistant to antibiotics (2LHF, 2JMM)" with "3.1 Proteins implicated in antibiotic resistance (OprH and OmpA)"

- line 254 - replace "presence of cavitation" with "presence of a polar cavity"

- line 254 - replace "The site of complexation of the second molecule is often associated with the local exposure of hydrophobicity on the surface" with "Secondary binding site for substrates are often associated with a hydrophobic region on the protein surface."

- line 258 - replace "Antifreeze proteins for example do show similar characteristics to those proteins discussed here. Antifreeze proteins do not require interaction with other molecules except water" with "Antifreeze proteins, which only interact with water, show similar characteristics to those proteins discussed here."

- line 262 - I am not sure what the authors are trying to say here. I would suggest replacing "Interpretation of the results presented here indicates low possibilities of interaction with molecules and, as the experiments show – the high resistance to antibiotics. This does not mean that the protein plays a role in the interaction with a specific molecule, which is probably passed" with "The distribution of polar and non-polar residues in OprH and OmpA, suggest that they interact weakly with substrate molecules and they therefore facilitate rapid transport of solutes between the periplasm and extracellular fluid. The analysis indicates that these proteins are likely to play a role in antibiotic resistance, consistent with previously published experimental evidence." 

- line 278 - replace "3.2. Oil transport (6QAM, 6QWR)" with "3.2 Oil transport (AlkL)

- line 279 - italics for "Pseudomonas oleovorans"

- line 283 - replace "The proteins represent identical sequences that additionally support the investigation of the environment’s influence on a structure" with "Different hydrophobic environments (lipid and detergent) have been shown to influence membrane protein structure (Cross, Murray, Watts (2013) European Journal of Biophysics 42(10):731-55) and AlkL provides an opportunity to test the application of the FOD-M analysis to two structures obtained using different NMR methodologies (Schubeis et al. 2020)."

- line 320 - replace "The structural differences visible in Figure 5 reveal the different status and composition of the out-barrel loops, the status of which in Table 2 is clearly different in the two compared structures" with "The large differences in the position of the out-barrel loops, visible in Figure 5, are reflected in the parameters calculated by this analysis, presented in Table 2."

- line 323 - replace "3.3. The barrels of higher diameter (1QJP, 4K3C, 6FSU, 4N75)" with "3.3. Beta barrel proteins of higher diameter (OmpA, BamA)"

- line 325 - add "Table 3" after "cross-section diameter than those discussed previously"

- line 345 - italics for "Pseudomonas aeruginosa"

- line 347 - replace "The distributions T and O in the case of 1QJP" with "The distributions T and O in the case of OmpA (PDB: 1QJP)"

- line 362 - replace "This is exactly the case with the 4N74 and 4K3C" with "This is exactly the case for the 4N74 and 4K3C structures of BamA."

- line 373 - "From the viewpoint of the analysis presented here, there is no obstacle to the transport of antibiotics through these channels in the proteins 6FSU, 4K3C, and 4N75 (based on the hydrophobicity distribution characteristics)" with "The analysis presented here indicates that, based on the hydrophobicity distribution characteristics, there is no barrier to the transport of antibiotics through BamA (PDB: 6FSU, 4K3C, and 4N75).

- line 376 - replace "3.4 Experimental modification - 4RLC, 4RL9 I 4RLB proteins" with "3.4 Experimentally modified proteins (OprF, CarO1 and CarO2)

- line 377 - replace "The 4RL9, 4RLB, and 4RLC proteins are discussed together..." with "The 4RL9, 4RLB, and 4RLC structures are discussed together..."

- line 381 - replace "Proteins with ID 4RL9, 4RLB and 4RLC..." with "Proteins Oprf (PDB 4RLC), CarO1 (PDB 4RL9) and CarO2 (PDB 4RLC)...

- line 393 - replace "accordant" with "consistent"

- line 394 - replace "The status of the out-barrel domain in the case of 4RLB is extremely different from the standard T distribution" with "The out-barrel domain of CarO2 (PDB 4RLB) differs dramatically from the standard T distribution"

- line 408 - replace "3.5 Auto-transporters (3AEH + 2QOM)" with "Auto-transporters (Hbp and EspP)"

- line 409 - replace "The next group analyzed are proteins called auto-transporters with codes PD 3AEH and 2QOM, respectively. They are self-sufficient in transporting the passenger domain through the outer membrane" with "Autotransporters such as Hbp (PDB 3AEH) and EspP (PDB 2QOM), have an N-terminal "passenger" domain which they can move through their central pore."

- line 411 - replace "The phenomenon of auto-transport has no explanation and no appropriate model. The process is associated with the secretion of the N-terminal domain referred to as "passenger". The beta-barrel shows proteolytic activity by digesting the N-terminal domain ("passenger")" with "The exact mechanism of auto-transport is unclear, however the beta barrel domain has been shown to have proteolysis activity."

- line 417 - italicize "E. coli"

- Table 3 - replace "PROTEIN" with "STRUCTURE" and add a separate column containing the name of each protein. 

- Table 3 - please rethink the position of this table - it might be better if it was placed under section 3.3.

- line 459 - replace "3.7. Transmembrane based on the helical bundle (5AZO)" with "3.7 A transmembrane protein with a helical bundle (PAO1)"

- line 460 - replace "The only representative of the transmembrane protein with the part anchored in the membrane being a helical structure is the protein with the code 5AZO" with "The PAO1 protein from Pseudomonas aeruginosa is a multidrug efflux pump and is anchored in the membrane by a helical bundle."

- line 478 - replace "It is difficult to conclude about the specificity of the system represented by this protein (especially with regard to its resistance to antibiotics) without the complete complex structure available" with "It is difficult to draw specific conclusions from the 5AZO structure about the role of PAO1 in antibiotic resistance, since the structure is not of the complete protein complex."

Author Response

REVIEWER I

Dear Reviewer I

Many thanks for the work you engaged in reviewing our paper.

We appreciate your efford to make our paper suitable for publication in Membranes.

We followed your suggestsions except two of tchem, where – according to our interpretation – the intention got changed.

The detailed comments are given below

Many thanks for your suggestions

Sincerely yours:

Irena Roterman

Comments and Suggestions for Authors

The authors have asked a native English speaker to check through their manuscript and to improve the quality of the writing. Unfortunately only very minor changes (mostly spelling) have been made and no serious attempt has been made to improve the grammar and the clarity of large sections. Here are some further suggested amendments.

I would urge the authors to ask a second native English speaker to work on the manuscript to improve the quality of the writing. At present the manuscript does not do justice to the scientific work being presented. 

The second native speaker was engaged to review the text.

IMPORTANT - a PDB code (e.g. 2LHF or 2JMM) indicates a STRUCTURE (from X-ray crystallography or NMR) and is not the same as the name of a protein (e.g. OprH or OmpA)

Many thanks for this comment. We corrected these expressions.

IMPORTANT - please make sure that all Latin organism names are italicized

We checked the text as to the latin names.

- line 17 - replace "Membrane proteins represent the opposite system, exposing hydrophobic residues on the surface and, in the case of ion channels, the polar residues are concentrated in the central part" with ""Membrane proteins represent the opposite system, exposing hydrophobic residues on the surface and, in the case of ion channels, the polar residues face inwards towards a central pore."

The sentences corrected.

- line 19 - replace "Their hydrophobicity distribution can be expressed using the complementary function to Gaussian (3DG) in the form 1-3DG" with "The distribution of hydrophobic residues in membrane proteins can be characterised using 1-3DG, a modified 3D Gauss function."

The word „complementary” shall stay as this expresses the mathematical expression which has been introduced.

- line 23 - replace "The identified differentiation translates into the specificity of the biological activity of these proteins, such as the type of molecule being transported and the resistance to antibiotics" with "The biological activity and substrate-specificity of these proteins, is determined by the distribution of polar and non-polar amino acids"

CORRECTED

- line 25 - replace "Comparative results help to identify the degree of resistance to antibiotics based on the possibility of interaction of a given protein with molecules resulting from the structure of the hydrophobic core" with "The analysis allows us to compare the ways in which the different proteins interact with antibiotics and helps us to understand their relative importance for development of resistance mechanisms."

CORRECTED

- line 27 - I am not sure what is meant here. It might be better to replace "Centrally located hydrophobic core introducing stabilization within the molecule is treated as a factor that prevents additional interactions with other molecules" with "We show that beta barrel membrane proteins with a hydrophobic core interact less strongly with the molecules that they transport." But I do not really understand what the authors are trying to say in this sentence.

The protein molecule with the hydrophobicity distribution accordant with idealised one (T) is not interested in interacting with other molecules but water due to hydrophilic surface. We publisehd many papers discussing this subject. This is why the presence of highly ordered hydrophobicity in beta-barrel molecule seems to prevent the interaction with any other molecules including antibiotics in particular.

- line 36 - replace "knowledge of their structures and functions appears important" with "knowledge of their structures and functions is important".

CORRECTED

- line 39 - replace "New solutions to overcome resistance, which commonly arises from the misuse and over-prescription of antibiotics, require the development of new solutions" with "New solutions to overcome resistance, which commonly arises from the misuse and over-prescription of antibiotics, are required."

CORRECTED

- line 50 - replace "The analysis of this phenomenon at the molecular level focuses on the stage of the mechanism of penetration of the designed antibiotic into the bacterial canal" with "The analysis of this phenomenon at the molecular level focuses on the mechanism of penetration of the designed antibiotic into transmembrane channels."

- line 53 - replace "These are proteins located in the outer membrane of bacteria that 53 connect the periplasm to the outside world" with "These proteins are located in the outer membrane of bacteria and connect the periplasm to the outside world."

CORRECTED

- line 69 - check spelling of "heliacal"

CORRECTED

- line 96 - replace "The FOD model assumes the possibility of describing the hydrophobicity distribution in globular proteins using a 3D Gaussian function" with "The FOD model uses a 3D Gaussian function to describe the distribution of hydrophobic amino acid residues in globular proteins."

CORRECTED

- line 114 - correct "where P the analyzed <<< missing words after “P”"

CORRECTED

- line 199 - replace "The summary of the obtained results for the description of the status of the discussed membrane proteins is presented in Table 2" with "Table 2 summarizes the results of the FOD-M analysis for members of the OmpX beta barrel membrane protein family."

CORRECTED

- line 211 - replace "The K values determine the degree of participation of a factor other than polar in shaping the structure" with "The K values indicate the relative importance of polar and non-polar residues in determining the protein structure."

The K value measures the participation of any other factors besides water molecules.

- line 211 and 220 - duplication?

REMOVED

- line 225 - replace "3.1. Proteins resistant to antibiotics (2LHF, 2JMM)" with "3.1 Proteins implicated in antibiotic resistance (OprH and OmpA)"

CHANGED

- line 254 - replace "presence of cavitation" with "presence of a polar cavity"

CORRECTED

- line 254 - replace "The site of complexation of the second molecule is often associated with the local exposure of hydrophobicity on the surface" with "Secondary binding site for substrates are often associated with a hydrophobic region on the protein surface."

CORRECTED

- line 258 - replace "Antifreeze proteins for example do show similar characteristics to those proteins discussed here. Antifreeze proteins do not require interaction with other molecules except water" with "Antifreeze proteins, which only interact with water, show similar characteristics to those proteins discussed here."

CORRECTED

- line 262 - I am not sure what the authors are trying to say here. I would suggest replacing "Interpretation of the results presented here indicates low possibilities of interaction with molecules and, as the experiments show – the high resistance to antibiotics. This does not mean that the protein plays a role in the interaction with a specific molecule, which is probably passed" with "The distribution of polar and non-polar residues in OprH and OmpA, suggest that they interact weakly with substrate molecules and they therefore facilitate rapid transport of solutes between the periplasm and extracellular fluid. The analysis indicates that these proteins are likely to play a role in antibiotic resistance, consistent with previously published experimental evidence." 

The interpretation is as follows:

The protein of hydrophobicity distribution accordant with the idealised one (T) is not able to interact with any other molecule except water or ions due to the polar surface. This is why this protein is not able to interact with molecules including antibiotics in particular.

- line 278 - replace "3.2. Oil transport (6QAM, 6QWR)" with "3.2 Oil transport (AlkL)

CORRECTED

- line 279 - italics for "Pseudomonas oleovorans"

CORRECTED

- line 283 - replace "The proteins represent identical sequences that additionally support the investigation of the environment’s influence on a structure" with "Different hydrophobic environments (lipid and detergent) have been shown to influence membrane protein structure (Cross, Murray, Watts (2013) European Journal of Biophysics 42(10):731-55) and AlkL provides an opportunity to test the application of the FOD-M analysis to two structures obtained using different NMR methodologies (Schubeis et al. 2020)."

CORRECTED ; the reference SUPPLEMENTED

- line 320 - replace "The structural differences visible in Figure 5 reveal the different status and composition of the out-barrel loops, the status of which in Table 2 is clearly different in the two compared structures" with "The large differences in the position of the out-barrel loops, visible in Figure 5, are reflected in the parameters calculated by this analysis, presented in Table 2."

CORRECTED

- line 323 - replace "3.3. The barrels of higher diameter (1QJP, 4K3C, 6FSU, 4N75)" with "3.3. Beta barrel proteins of higher diameter (OmpA, BamA)"

CORRECTED

- line 325 - add "Table 3" after "cross-section diameter than those discussed previously"

CORRECTED

- line 345 - italics for "Pseudomonas aeruginosa"

CORRECTED

- line 347 - replace "The distributions T and O in the case of 1QJP" with "The distributions T and O in the case of OmpA (PDB: 1QJP)"

CORRECTED

- line 362 - replace "This is exactly the case with the 4N74 and 4K3C" with "This is exactly the case for the 4N74 and 4K3C structures of BamA."

CORRECTED

- line 373 - "From the viewpoint of the analysis presented here, there is no obstacle to the transport of antibiotics through these channels in the proteins 6FSU, 4K3C, and 4N75 (based on the hydrophobicity distribution characteristics)" with "The analysis presented here indicates that, based on the hydrophobicity distribution characteristics, there is no barrier to the transport of antibiotics through BamA (PDB: 6FSU, 4K3C, and 4N75).

CORRECTED

- line 376 - replace "3.4 Experimental modification - 4RLC, 4RL9 I 4RLB proteins" with "3.4 Experimentally modified proteins (OprF, CarO1 and CarO2)

CORRECTED

- line 377 - replace "The 4RL9, 4RLB, and 4RLC proteins are discussed together..." with "The 4RL9, 4RLB, and 4RLC structures are discussed together..."

CORRECTED

- line 381 - replace "Proteins with ID 4RL9, 4RLB and 4RLC..." with "Proteins Oprf (PDB 4RLC), CarO1 (PDB 4RL9) and CarO2 (PDB 4RLC)...

CORRECTED

- line 393 - replace "accordant" with "consistent"

CORRECTED

- line 394 - replace "The status of the out-barrel domain in the case of 4RLB is extremely different from the standard T distribution" with "The out-barrel domain of CarO2 (PDB 4RLB) differs dramatically from the standard T distribution"

CORRECTED

- line 408 - replace "3.5 Auto-transporters (3AEH + 2QOM)" with "Auto-transporters (Hbp and EspP)"

CORRECTED

- line 409 - replace "The next group analyzed are proteins called auto-transporters with codes PD 3AEH and 2QOM, respectively. They are self-sufficient in transporting the passenger domain through the outer membrane" with "Autotransporters such as Hbp (PDB 3AEH) and EspP (PDB 2QOM), have an N-terminal "passenger" domain which they can move through their central pore."

CORRECTED

- line 411 - replace "The phenomenon of auto-transport has no explanation and no appropriate model. The process is associated with the secretion of the N-terminal domain referred to as "passenger". The beta-barrel shows proteolytic activity by digesting the N-terminal domain ("passenger")" with "The exact mechanism of auto-transport is unclear, however the beta barrel domain has been shown to have proteolysis activity."

CORRECTED

- line 417 - italicize "E. coli"

CORRECTED

- Table 3 - replace "PROTEIN" with "STRUCTURE" and add a separate column containing the name of each protein. 

THE NEW COLUMN ADDED

- Table 3 - please rethink the position of this table - it might be better if it was placed under section 3.3.

TABLE 3 MOVED to SECTION 3.3

- line 459 - replace "3.7. Transmembrane based on the helical bundle (5AZO)" with "3.7 A transmembrane protein with a helical bundle (PAO1)"

CORRECTED

- line 460 - replace "The only representative of the transmembrane protein with the part anchored in the membrane being a helical structure is the protein with the code 5AZO" with "The PAO1 protein from Pseudomonas aeruginosa is a multidrug efflux pump and is anchored in the membrane by a helical bundle."

CORRECTED

- line 478 - replace "It is difficult to conclude about the specificity of the system represented by this protein (especially with regard to its resistance to antibiotics) without the complete complex structure available" with "It is difficult to draw specific conclusions from the 5AZO structure about the role of PAO1 in antibiotic resistance, since the structure is not of the complete protein complex."

CORRECTED

Round 4

Reviewer 1 Report

The manuscript is much improved and the writing is considerably clearer. I have a few minor suggests for the authors below.

- line 17 - replace "Membrane proteins represent the opposite system, exposing hydrophobic residues on the surface" with "In contrast, membrane proteins have exposed hydrophobic residues on the surface"

- line 41 - delete extra space between "poses" and "a new challenge"

- line 47 - delete extra space between "essential" and "when endeavoring"

- line 222 - replace "which can be interpreted as structures with a centric hydrophobic core and polar residue exposure" with "which can be interpreted as structures with a central hydrophobic core and exposed polar residues"

- line 240 - missing "." after "Pseudomonas aeruginosa"

- line 273 - replace "Two structures, one soluted in the presence of lipid" with "Two structures, one solved in the presence of lipid"

- line 303 - replace "for the two structural forms of the 6QWR lipid and the 6QAM detergent" with "for the two structures of AlkL (6QWR in lipid and 6QAM in detergent)"

- line 312 - replace "3D presentation revealing the diversity of the discussed structures. A - 6QAM – detergent environment (gray)" with "Comparison of AlkL structures: A - 6QAM – detergent environment (gray)"

- line 321 - I am not sure what is meant here. I suggest replacing "This reveals a differentiation consisting in a gradual increase in the inconsistency of the hydrophobicity distribution from a system close to a hydrophobic core with a low K value to a system that deviates significantly from distribution T, thereby revealing the importance of high K values" with "There is considerable variation in the hydrophobicity profiles of members of this group, as shown in the differences in the K values calculated in this analysis. The 1QJP structure of OmpA (average internal diameter= 15.9 Å; K= 0.4) closely resembles other beta barrel proteins with a clear hydrophobic core, whereas the 4K3C structure of BamA (average internal diameter= 35.2 Å; K= 1.2) has a more polar interior."

- line 379 - replace "no barrier exists to the transport of antibiotics through the BamA (PDB : 6FSU, 4K3C, and 4N75)" with either "no barrier exists to the transport of antibiotics through the BamA pore (PDB : 6FSU, 4K3C, and 4N75)" or "no barrier exists to the transport of antibiotics through BamA (PDB : 6FSU, 4K3C, and 4N75)"

- line 486 - delete extra space between "proteins" and "active"

- line 495 - delete extra comma after "particular"

- line 500 - start a new paragraph before "Numerous bacterial species"

- line 516 - replace "that takes into account" with "that take into account"

- line 519 - replace "the varying degrees to which the hydrophobic environment shapes the structure of the discussed proteins, despite the common environment of their activity" with "the varying degrees to which the hydrophobic environment shapes their structures."

Author Response

REVIEWER I

Dear Reviewer I

Many thanks for your work done by you to make our paper of higher quality.

We followed your advices.

We hope – you find tchem acceptable.

Sincerely Yours;

Irena Roterman

Comments and Suggestions for Authors

The manuscript is much improved and the writing is considerably clearer. I have a few minor suggests for the authors below.

- line 17 - replace "Membrane proteins represent the opposite system, exposing hydrophobic residues on the surface" with "In contrast, membrane proteins have exposed hydrophobic residues on the surface"

CORRECTED

- line 41 - delete extra space between "poses" and "a new challenge"

CORRECTED

- line 47 - delete extra space between "essential" and "when endeavoring"

CORRECTED

- line 222 - replace "which can be interpreted as structures with a centric hydrophobic core and polar residue exposure" with "which can be interpreted as structures with a central hydrophobic core and exposed polar residues"

CORRECTED

- line 240 - missing "." after "Pseudomonas aeruginosa"

CORRECTED

- line 273 - replace "Two structures, one soluted in the presence of lipid" with "Two structures, one solved in the presence of lipid"

CORRECTED

- line 303 - replace "for the two structural forms of the 6QWR lipid and the 6QAM detergent" with "for the two structures of AlkL (6QWR in lipid and 6QAM in detergent)"

CORRECTED

- line 312 - replace "3D presentation revealing the diversity of the discussed structures. A - 6QAM – detergent environment (gray)" with "Comparison of AlkL structures: A - 6QAM – detergent environment (gray)"

CORRECTED

- line 321 - I am not sure what is meant here. I suggest replacing "This reveals a differentiation consisting in a gradual increase in the inconsistency of the hydrophobicity distribution from a system close to a hydrophobic core with a low K value to a system that deviates significantly from distribution T, thereby revealing the importance of high K values" with "There is considerable variation in the hydrophobicity profiles of members of this group, as shown in the differences in the K values calculated in this analysis. The 1QJP structure of OmpA (average internal diameter= 15.9 Å; K= 0.4) closely resembles other beta barrel proteins with a clear hydrophobic core, whereas the 4K3C structure of BamA (average internal diameter= 35.2 Å; K= 1.2) has a more polar interior."

CORRECTED

- line 379 - replace "no barrier exists to the transport of antibiotics through the BamA (PDB : 6FSU, 4K3C, and 4N75)" with either "no barrier exists to the transport of antibiotics through the BamA pore (PDB : 6FSU, 4K3C, and 4N75)" or "no barrier exists to the transport of antibiotics through BamA (PDB : 6FSU, 4K3C, and 4N75)"

CORRECTED

- line 486 - delete extra space between "proteins" and "active"

CORRECTED

- line 495 - delete extra comma after "particular"

CORRECTED

- line 500 - start a new paragraph before "Numerous bacterial species"

CORRECTED

- line 516 - replace "that takes into account" with "that take into account"

CORRECTED

- line 519 - replace "the varying degrees to which the hydrophobic environment shapes the structure of the discussed proteins, despite the common environment of their activity" with "the varying degrees to which the hydrophobic environment shapes their structures."

CORRECTED